



# Interactions between the stratospheric polar vortex and Atlantic circulation on seasonal to multi-decadal timescales

Oscar Dimdore-Miles[1], Lesley Gray[1,2], Scott Osprey[1,2], Jon Robson[2,3], Rowan Sutton[2,3], and Bablu Sinha[4]

[1]Atmospheric, Oceanic and Planetary Physics, Department of Physics, University of Oxford, OX1 3PU, UK
[2]National Centre for Atmospheric Science
[3]Department of Meteorology, University of Reading, Reading, RG6 6ET
[4]National Oceanography European Way, Southampton SO14 3ZH

**Correspondence:** Oscar Dimdore-Miles (oscar.dimdore-miles@physics.ox.ac.uk)

**Abstract.**

Variations in the strength of the Northern Hemisphere winter polar stratospheric vortex can influence surface variability in the Atlantic sector. Disruptions of the vortex, known as sudden stratospheric warmings (SSW), are associated with an equatorward shift and deceleration of the North Atlantic jet stream, negative phases of the North Atlantic Oscillation as well as cold snaps over Eurasia and North America. Despite clear influences at the surface on sub-seasonal timescales, how stratospheric vortex variability interacts with ocean circulation on decadal to multi-decadal timescales is less well understood. In this study, we use a 1000-year pre-industrial control simulation of the UK Earth System Model to study such interactions using a wavelet analysis technique to examine non-stationary periodic signals in the vortex and ocean. We find that intervals which exhibit persistent anomalous vortex behaviour lead to oscillatory responses in the Atlantic Meridional Overturning Circulation (AMOC). The origin of these responses appears to be highly non-stationary with spectral power in vortex variability and the AMOC at periods of 30 and 50 years. In contrast, AMOC variations on longer timescales (near 90-year periods) are found to lead to a vortex response, through a pathway involving the equatorial Pacific and Quasi-biennial Oscillation. Using the relationship between persistent vortex behaviour and the AMOC response established in the model, we use a regression analysis to estimate the potential contribution of the 8 year SSW hiatus interval in the 1990s to the recent negative trend in AMOC observations. The result suggests that approximately 30% of the trend may have been caused by the SSW hiatus.



# 1 Introduction

Variations in the strength of the Northern Hemisphere (NH) stratospheric polar vortex associated with sudden stratospheric warming (SSW) events is the single largest source of interannual variability in the NH winter stratosphere. Extreme disruptions of the vortex during SSWs also represent a key mechanism for stratosphere-troposphere coupling and are widely acknowledged
to lead to anomalies in NH mid-latitude surface climate, particularly in the North Atlantic sector (Baldwin and Dunkerton, 2001). Equally, the absence of an SSW, when the vortex is relatively undisturbed by waves propagating from the troposphere so that the vortex is unusually strong and cold, also has significant impacts on surface weather (Shaw and Perlwitz, 2013; Lawrence et al., 2020). An understanding of SSW dynamics and vortex variability is therefore important for seasonal to sub-seasonal forecasting of surface weather (Domeisen et al., 2020a, b) as they provide a significant source of predictive skill.
The majority of studies that have examined the associations between the stratospheric polar vortex and surface anomalies have considered their in-season impact. However, the NH winter stratosphere also exhibits variability on decadal to multi-decadal timescales (Dimdore-Miles et al., 2021). The interaction between this low-frequency variability in vortex strength and variations in surface climate on similar timescales is not fully understood.

The impact of stratospheric polar vortex (hereafter referred to as 'the vortex') variations on surface climate variability was
highlighted in the seminal work of Baldwin and Dunkerton (2001) which demonstrates a steady downward propagation of Northern Annular Mode (NAM) anomalies from the middle stratosphere to the surface following strong and weak vortex events in the ERA40 reanalysis dataset. The anomalies associated with a weak (strong) vortex were shown to persist at the surface intermittently for approximately 60 days after the event and projected significantly onto the negative (positive) pattern of the North Atlantic Oscillation (NAO). Subsequent modelling and observational studies have corroborated these results:
Domeisen (2019) show, using ERA40 and ERA-Interim datasets, that approximately two-thirds of SSW events are followed either by a switch from positive to negative NAO or a persistent negative NAO pattern. Charlton-Perez et al. (2018) consider this coupling from the perspective of tropospheric weather regimes and find, in both ERA-Interim and the ECMWF Integrated forecast model, a 40-60% increase in probability of transition to a negative phase of the NAO given a one standard deviation reduction in polar vortex strength. This SSW influence on the NAO is relatively well represented in GCMs (Baldwin et al.,
2021) and idealised modelling studies are also able to show a direct downward influence of events on the NAO (White et al., 2020; Gerber et al., 2009), although some studies have noted that simple models tend to over-represent the persistence of surface anomalies (Gerber et al., 2008a, b),

The influence of SSW events on negative NAO phase probability has subsequently been shown to influence other features of NH mid-latitude climate. Thompson et al. (2002) show in NCEP–NCAR reanalysis that SSWs are followed by a 60 day
interval of anomalously low surface temperatures in eastern North America, northern Europe, and eastern Asia. Subsequent observational studies find similar response patterns to SSWs (Kolstad et al., 2010; King et al., 2019; Lehtonen and Karpechko, 2016) and the effect can also be seen in GCM simulations (Tomassini et al., 2012; Lehtonen and Karpechko, 2016). Further impacts of SSWs on tropospheric circulation include an equatorward shift and deceleration of the North Atlantic eddy-driven jet stream (Hitchcock and Simpson, 2014; Maycock et al., 2020). Links with persistent blocking events are also shown in





observations and modelling studies (Davini et al., 2014; Vial et al., 2013) although Taguchi (2008) finds no significant link between the phenomena.

While the in-season influence of SSWs on tropospheric circulation and surface variability is discussed extensively in previous work, their coupling with modes on longer timescales such as ocean variability is less well understood. One of the primary features of Atlantic Ocean variability is the Atlantic Meridional Overturning Circulation (AMOC) which consists of a north-

ward transfer of warm, saline water that occurs in the top 2km of the Atlantic Ocean (the "upper cell") accompanied by a corresponding return flow of southward transport at lower depths (the "lower cell") (Kuhlbrodt et al., 2007; Xu et al., 2014; Buckley and Marshall, 2016). The strength of the AMOC varies significantly on decadal-centennial timescales (Delworth et al., 1993; Biastoch et al., 2008; Tulloch and Marshall, 2012; Menary et al., 2012) and is thought to be a key driver of North American and European surface variability via modulation of Atlantic sea surface temperatures (SSTs) and heat transport (Knight

et al., 2005; Delworth and Mann, 2000; Frierson et al., 2013; Frankignoul et al., 2013). It has been shown to influence a diverse range of features such as European summertime temperatures (Sutton and Hodson, 2005), biogeochemical conditions in the northwest Atlantic (Lavoie et al., 2019) as well as abrupt climate shifts in paleoclimate records (Alley, 2007; Cheng et al., 2009).

Multiple potential drivers of AMOC variability at different timescales have been studied extensively in both observations and

modelling studies. On intra-annual to inter-annual timescales, variability in the AMOC has been closely associated with wind variations over the North Atlantic region through Ekman-transport anomalies or wind stress curl forcing (Wang et al., 2019; McCarthy et al., 2012; Mielke et al., 2013; Yang, 2015). On inter-annual to decadal timescales, AMOC variability has been associated with buoyancy anomalies in the subpolar region, particularly in the Labrador Sea (Delworth et al., 1993; Medhaug et al., 2012). This mechanism is linked to variability in mixed layer depth and the occurrence of deep convection over the

same region, particularly in NH winter (Böning et al., 2006; Biastoch et al., 2008; Robson et al., 2012; Wang et al., 2015). Mixed-layer anomalies in the Labrador Sea are an indication of the strength of deep convection in this region, which has been shown to be associated with AMOC variations in modelling studies (Eden and Willebrand, 2001; Eden and Jung, 2001) and observations (Latif and Keenlyside, 2011).

An association between stratospheric polar vortex variability and the AMOC on decadal timescales has been previously

investigated (Reichler et al., 2012; Schimanke et al., 2011) but the mechanism of its influence remains unclear. For example, Reichler et al. (2012) examine the response of the AMOC to strong and weak polar vortex events and show a lagged, oscillatory response in the AMOC. They propose a pathway involving alterations of wind stress and ocean-atmosphere heat flux anomalies in the West Atlantic due to the changed NAO patterns following the vortex events. The effect is prominent in a pre-industrial (PI) control of a single model (GFDL-CM2.1) and to some extent in a suite of CMIP5 models. An impact of long-term changes

in the NAO on the strength of the AMOC is supported by a number of studies (Visbeck et al., 1998; Delworth and Dixon, 2000; Delworth and Greatbatch, 2000; Eden and Willebrand, 2001; Lohmann et al., 2009; Robson et al., 2012). Most recently Delworth and Zeng (2016) used a set of idealised GCM experiments in which they impose a perpetual ocean-atmosphere heat flux pattern associated with different NAO phases. They find significantly different AMOC mean states depending on the imposed pattern (a stronger AMOC under positive NAO flux conditions than a control simulation). Haase et al. (2018) also



analysed the in-season influence of SSW events on the NAO and ocean-atmosphere heat fluxes that then impacts the strength of deep convection in the North Atlantic, using the CESM1 WACCM model. The study notes the presence of an anomalously shallow mixed layer depth in the Labrador Sea following an SSW event.

A key result from Reichler et al. (2012) is the decadal modulation of the SSW-AMOC co-variability. However, decadal to multi-decadal variability in vortex strength is not well understood. Some studies have focused on potential polar vortex impacts

at the surface but suffer from low statistical significance due to the short observational record. Garfinkel et al. (2017), Garfinkel et al. (2015) and Cohen et al. (2009) link decadal fluctuations in vortex strength with modulation of the global warming signal in Eurasian surface temperature in both reanalyses and CMIP3 models. Schimanke et al. (2011) demonstrate multi-decadal signals in SSW occurrence in a multi-century GCM simulation and propose an influence of these signals on similar period variability in Eurasian snow cover and Atlantic SSTs. However, results from this study are difficult to interpret as the GCM

used (EGMAM: ECHO-G with Middle Atmosphere Model) exhibits significant bias in its vortex representation with a mean SSW rate of only 2 events per decade compared to 6 events per decade in most reanalyses (Ayarzagüena et al., 2019). They note that repeating their study with a more advanced model is required to corroborate their findings. Manzini et al. (2012) examined decadal fluctuations in SSW events in a 260 year prescribed SST simulation of a GCM and analyse their impacts at the surface. They show that decadal vortex variability excites similar timescale variations in surface temperature and sea ice

coverage between Greenland and Norway over the Atlantic sector. They propose this connection to be indicative of a delayed response of the AMOC to stratospheric forcing via the NAO which subsequently influences northward Atlantic heat transfer and sea ice melt rates as well as surface temperatures anomalies.

More recently, Dimdore-Miles et al. (2021) (henceforth referred to as DM21) examined long-term variability in the strength of the winter polar stratospheric vortex in a 1000-yr PI control simulation of the UK Earth System Model. They identified

sequences of up to 11 consecutive years in which at least one SSW occurred every year (similar to that observed during the period 1998-2004) and also sequences of up to 12 consecutive years with a strong undisturbed vortex as was observed during the 1990s (Manney et al., 2005; Pawson and Naujokat, 1999). They identified multi-decadal signals of 90-year periodicity in the latter that persisted for approximately 450 years of the 1000 years. Using wavelet and cross-spectral analysis, they associated this with a similar signal in the amplitude modulation of the Quasi Biennial Oscillation (QBO) and proposed that the vortex

variability was driven by the QBO through modulation of the Holton-Tan relationship (Lu et al., 2008, 2014). However, the focus of that paper was primarily in the stratosphere, identifying and understanding the relationship between the long-term QBO and vortex variability. In this paper, we use the same 1000-yr pi-control simulation to extend that study to examine the links between long-term vortex variability and the North Atlantic including oceanic modes. We first examine the near-surface in-season response to extreme vortex events to demonstrate that the model is able to reproduce corresponding anomalies in

mean sea level pressure (MSLP), ocean-atmosphere heat flux and SSTs consistent with previous studies. We then explore the response on multi-decadal timescales using a wavelet analysis method to examine non-stationary signals in the vortex and the AMOC variability. We find that multi-year intervals which exhibit the same type of persistent, anomalous vortex behaviour also exhibit co-variability with the AMOC across multiple timescales. The interactions and feedback mechanisms on these different timescales are explored in more detail using a combination of lag/lead composite analysis as well as wavelet spectral





decomposition method. The amplitude and lag of the AMOC response to a multi-year period of strong polar vortex in the model is determined. This is then used to estimate the potential contribution of the observed 8 consecutive years with a very stable, undisturbed stratospheric vortex in the 1990s (Pawson and Naujokat, 1999) to the recent observed negative trend in the strength of the AMOC. The paper is structured as follows: Section 2 describes the GCM used in the investigation, the spectral analysis method (wavelet analysis) and the relevant climate indices. Section 3 presents results from the analysis. Section 4

provides a summary and discussion of the results.

## 2  Data and Methods

### 2.1  Model Configuration

This work utilises the same model as that analysed in DM21, The first version of the UK Earth System Model (henceforth referred to as UKESM). UKESM is a stratosphere resolving coupled ocean-atmosphere-land-sea ice model. It contains 85

vertical levels in the atmospheric domain simulated by the Global Atmosphere 7.1 component (GA7.1) 35 of which lie above 35km in altitude (Walters et al., 2019; Williams et al., 2018). GA7.1 runs at N96 horizontal resolution (~135km at the equator). Ocean circulation is simulated by GO6.0 (Storkey et al., 2018) which contains 75 vertical levels and runs at $1°$ horizontal resolution. Additional interactive components simulating land surface, sea ice and atmospheric chemistry processes are added via coupling with JULES (GL7.0), CICE (GSI8.1) and UK Chemistry and Aerosols (UKCA) models respectively (Walters

et al., 2019; Ridley et al., 2018; Mulcahy et al., 2018). The model's representation of the North Atlantic Climate system has been examined previously by Robson et al. (2020); this study shows that UKESM is able to realistically simulate key features such as the AMOC. However, this study highlights the model's under-representation of variability in the AMOC, an issue which is reported in other physical models (Roberts et al., 2014).

We utilise the same 1000 year PI control simulation as examined in DM21. This simulation is spun-up to achieve model

equilibrium before initialising following the method outlined in Yool et al. (2020). The run is forced using CMIP6 pre-industrial values for concentrations of major GHGs (global mean 284.317ppm $CO_2$, 808.25ppb $CH_4$, 273.02ppb $N_2O$). There are no volcanic eruptions in the simulation but a background stratospheric volcanic aerosol level is imposed using climatological values between 1850 and 2014 estimated from satellite products and other model simulations (Menary et al., 2018). The simulation does not exhibit a solar cycle. We choose to analyse a pi-control simulation due to the length of integration performed

(1000 years) compared to the timescales of stratospheric variations shown in DM21 (near 90-year variability). This length of simulation provides a greater number of possible cycles of such variability available for analysis compared to the use of a historical simulation. Furthermore, a pi-control allows us to analyse the internal variability of the stratosphere.

To estimate the contribution of stratospheric variations to recent observed AMOC trends we also make use of observation-based datasets of the atmosphere and oceans. First, we utilise the reanalysis data from the European Centre for Medium-Range

Weather Forecasts (ECMWF): ERA5 (Hersbach et al., 2020) for assimilated observations for geopotential height (GPH) and MSLP fields downloaded from https://climate.copernicus.eu/climate-reanalysis (last accessed 10/08/21). Second, the Rapid Array Dataset which provides time-depth profiles for the meridional overturning mass streamfunction in the Atlantic region at





26°N (Moat et al., 2020). These data are measured through a combination of ocean mooring, ship-based, satellite and submarine telephone cable observations to estimate the strength of primary contributions to the meridional overturning circulation: Ekman transport (through wind stress), transport through the Florida Straits and transport driven by East-West density gradients between the American and African continents (McCarthy et al., 2015).

### 2.2 Wavelet Analysis

We employ the same wavelet analysis method as DM21 based on Torrence and Compo (1998) to examine potential non-stationary spectral characteristics of time series data over a range of periods. This method is outlined in full in section 2.3 of DM21 and is reproduced briefly here.

The wavelet transform of a 1-dimensional time series, $x$, of length $N$ and uniform timestep $\delta t$ is given by the convolution between the series and a wavelet function $\psi$ which has been scaled by the quantity $s$,

$$W_n(s) = \sum_{n'=0}^{N-1} x_{n'} \psi^* \left[ (n' - n) \frac{\delta t}{s} \right],\tag{1}$$

where $s$ is the scale of the wavelet which indicates its period and $n$ is the time index. Varying $s$ and translating along the time axis builds up a power spectrum for $x$ in the time-period domain given by $|W_n(s)|^2$. Following Torrence and Compo (1998) we vary the scale parameter in increasing powers of 2 such that $s_j = s_0 2^{j\delta j}$ and $j = 0, 1, ..., J$, where $j$ is the index for the wavelet scale, $s_0$ is the smallest resolvable scale for the series and $J$ is the index corresponding to the largest scale given by

$$J = \delta j^{-1} log_2 \left( \frac{N \delta t}{s_0} \right).\tag{2}$$

The translated and scaled wavelet evaluated at a given scale, $s$, has the form

$$\psi^* \left[ (n' - n) \frac{\delta t}{s} \right] = \left( \frac{\delta t}{s} \right)^{1/2} \psi_0 \left[ (n' - n) \frac{\delta t}{s} \right]\tag{3}$$

and we select the form of the wavelet basis function $\psi_0$ following the recommendation of Torrence and Compo (1998) as a Morlet wavelet, an oscillatory function enveloped by a Gaussian which is expressed as

$$\psi_0(p) = \pi^{-1/4} e^{i \omega_0 p} e^{\frac{p^2}{2}}.\tag{4}$$

The advantages of using a Morlet wavelet for analysing signals in climate time series is primarily due to its ability to resemble many of the features commonly observed in climate time series (e.g. changes in dominant period, and amplitudes). The full argument for this choice is presented in Lau and Weng (1995) as well as DM21. To directly compare spectra of different indices, we normalise all time series by subtracting the mean and dividing by its standard deviation before performing





the wavelet transform. In order to effectively compare spectral power across a range of frequencies we additionally scale the power spectrum by dividing by the scale parameter ($s_j$ defined above) associated with each frequency. This is done following

the methodology of Liu et al. (2007) which shows that un-scaled spectra exhibit a bias towards overestimated powers at longer periods and that an effective comparison across timescales is possible with such scaling. We also define a confidence interval for wavelet power observed for the series $x$ by comparing the observed power to that produced by a time series modelled as a first-order auto-regressive (AR1, red noise) process, $r$, given by

$$r_n = \alpha r_{n-1} + z_n, \tag{5}$$

where $\alpha$ is the lag-1 autocorrelation of $x$ and $z_n$ is a Gaussian white noise term. Torrence and Compo (1998) show that the power spectrum of this is $\chi^2$ distributed and therefore can be used to define a 95% confidence interval for any observed power.

### 2.3    Cross Wavelet Spectra

We also define a measure of coincident spectral power between two time series, the cross wavelet spectrum. This metric indicates whether two series exhibit power at the same timepoints and frequencies. The cross wavelet spectrum of two time

series $x$ and $y$ with associated wavelet spectra $W_n^x$ and $W_n^y$ is calculated by projecting one spectrum onto the other:

$$|W_n^{xy}(s)| = |W_n^{x*}(s)W_n^y(s)|, \tag{6}$$

where $W_n^{x*}(s)$ is the complex conjugate of the wavelet power spectrum of $x$ (Grinsted et al., 2004). The complex argument of $W_n^{xy}(s)$ gives the local phase difference between signals in $x$ and $y$ in frequency-time space. The phase relationship between the two time series can be represented by a vector that subtends an angle representing the phase difference in radians: On all

plots of cross spectra, arrows to the right (left) denotes signals in the two series which are in-phase and correlated (anti-correlated). Vertical arrows indicate a phase relationship of $\frac{\pi}{2}$ between the time series, so that the evolution of one is correlated with the time rate-of-change of the other. As for individual power spectra, we define a confidence interval for which cross power of a larger amplitude is deemed significant (>95% confidence interval) by comparing power exhibited by actual series with a theoretical red noise process. The cross power of two such AR1 processes is theoretically distributed such that the

probability of obtaining cross power greater than a set of red-noise processes is

$$D\left( \frac{|W_n^{xy}(s)|}{\sigma_x \sigma_y} < p \right) = \frac{Z_\nu(p)}{\nu} \sqrt{P_k^x P_k^y}, \tag{7}$$





where $\sigma$ denotes the standard deviation of the time series, $Z$ is the confidence interval defined by $p$ ($Z$ = 3.999 for 95% confidence), $\nu$ is the degrees of freedom for a real wavelet spectrum ($\nu$ = 2) and $P_k^x$ is the theoretical Fourier spectrum of the AR1 process. For a given wavenumber k and given by

$$P_k = \frac{1 - \alpha^2}{|1 - \alpha e^{2i\pi k}|^2}.$$
(8)

## 2.4 Model Diagnostics

We utilise the Northern Annular Mode (NAM) as a metric for the strength of the vortex as used by Baldwin and Dunkerton (2001) as well as numerous subsequent studies. The NAM is defined as the 1st principal component (PC) of the zonal mean, deseasonalised geopotential height field evaluated at latitudes north of $20°N$ over the NH winter season (Dec-Mar) on a given

pressure level. To measure the vortex strength we evaluate the NAM at 10hPa which is the pressure level used to identify major SSWs, and the resulting index is henceforth known as $NAM_{10}$. An individual vortex event (either strong or weak) is recorded when the daily $NAM_{10}$ crosses a threshold of +1.5 (strong) or -2 (weak). The day on which this reversal occurs is referred to as the central date. After this date, the $NAM_{10}$ must recover to westerly for at least 10 consecutive days (which is the approximate radiative timescale of the mid-stratosphere) before another event can be recorded. The strong threshold value

for events is chosen in accordance with the methodology of Baldwin and Dunkerton (2001) and the weak threshold selected such that it results in approximately the same rate of weak events (SSWs) as is reported in DM21 using the same simulation of UKESM but with a zonal wind definition of SSWs (0.54 events/winter).

We also use the $NAM_{10}$ to derive an index for the appearance of intervals of consecutive winters which show persistent vortex behaviour. The persistent $NAM_{10}$ interval index is defined as follows: First, the $NAM_{10}$ is averaged over each NH

winter season (December-March), this gives a measure of mean vortex strength for each winter. Second, this index is smoothed using a Gaussian filter which is carried out through a convolution of the time series with a 1D Gaussian kernel in the time domain given by

$$f(t,\sigma) = \frac{1}{\sqrt{2\pi\sigma^2}} e^{-\frac{1}{2}\left(\frac{t}{\sigma}\right)^2}$$
(9)

where $\sigma$ is the standard deviation of the distribution defined by the kernel. We choose $\sigma$ = 2 years following the method of

Reichler et al. (2012) and as a method analogous to the 5-year smoothing applied to an SSW time series in DM21. The selection of $\sigma = 2$ years allows contributions to the smoothed value from values approximately 7 years either side of the central year as the value of the Gaussian window decays to near 0 approximately $3.5\sigma$ from its mean. However, the largest contributions come from 3-4 years either side of the central year. This allows the smoothing to capture instances of ~6-8 consecutive years with persistent vortex behaviour, a similar length to intervals observed in reanalysis (e.g. the 1990s, Pawson and Naujokat (1999)).

We subsequently define persistent $NAM_{10}$ intervals, when the vortex exhibits the same type of behaviour for a number of consecutive years, using extreme values of the smoothed $NAM_{10}$ index. A persistent $NAM_{10}$ interval is recorded when the smoothed $NAM_{10}$ index value falls within the top 5 percentile values. Once such an interval occurs, another cannot be





recorded for 15 years after to avoid choosing multiple central years within the same interval. Using 5 percentile values gives
approximately the same rate of persistent vortex intervals as is reported in Reichler et al. (2012) so we proceed with this
threshold throughout for a direct comparison with this study. Tests were also carried out to assess the sensitivity of our results
to this threshold and are reported in section 3.

We define an AMOC index following the procedure in Reichler et al. (2012). The AMOC is defined using the overturning
streamfunction field in the Atlantic sector. At each time point, the AMOC index is the maximum streamfunction value at any
depth at a chosen latitude. We evaluate the index at 30N, 45N and 50N and measure the response and co-variability with the
$NAM_{10}$ time series and other climate indices defined below. We derive the observed AMOC index from the Rapid Array data
as the maximum MOC at each time point at 26N. We also utilise a definition of the North Atlantic Oscillation from Hurrell
et al. (2003). The NAO index is defined as the 1st PC of the Dec-Mar MSLP in the region $20° - 80°N, 90°W - 40°E$. The
PC is calculated by taking the first empirical orthogonal function (EOF) of deseasonalised MSLP anomalies and projecting
this EOF onto the anomaly field. We additionally derive an Ocean-Atmosphere heat flux field defined as the sum of latent and
sensible heat fluxes between the ocean surface and the atmosphere (i.e. positive values indicate the exchange of heat from the
ocean to the atmosphere). We derive an index for the occurrence of deep convection anomalies in the equatorial eastern Pacific
region. This index is defined by the top of atmosphere outgoing longwave radiation (OLR) averaged over the box 10° S–10° N,
240° –290° E. The OLR field is utilised as it acts as a proxy for the occurrence of convection anomalies - When deep convection
is enhanced, cloud top height is increased and therefore OLR is reduced. The East Pacific box is selected following a sensitivity
analysis to establish the region which exhibits 90-year timescales variations in OLR. It is also a similar region to studies that
consider east pacific ENSO patterns which is identified as a separate mode of variability to the traditionally used central pacific
ENSO region (Johnson, 2013).

We also utilise the same QBO metric as in DM21. This is defined as the Zonal Mean Zonal Wind (ZMZW) averaged over the
latitude range 5° S–5° N, and the pressure level range 15-30hPa. This metric captures the degree of vertical coherence in the
QBO, an attribute shown to be important for QBO teleconnections with the NH mid-latitudes (Andrews et al., 2019). Following
the method of DM21 we also define the instantaneous amplitude of the deep QBO, which exhibits significant modulation in
the westerly phase (see DM21 figures 10 and 11), using the Hilbert transform of the QBO ($Hil[QBO(t)]$) defined as

$$Hil[QBO(t)] = \frac{1}{\pi t} * QBO(t), \tag{10}$$

where $*$ signifies a convolution and $t$ is discretised time. The time varying Amplitude, $A(t)$, can then be expressed in terms
of $QBO(t)$ and $Hil[QBO(t)]$ such that

$$QBO(t) + Hil[QBO(t)]i = A(t)e^{i\theta}, \tag{11}$$

where $\theta(t)$ is the instantaneous phase angle - a measure of signal progression through a cycle at time $t$.

Finally, We analyse the relationship between the magnitude of smoothed stratospheric $NAM_{10}$ extremes and 17-year lagged
AMOC anomalies using linear regression with a single predictor (the lagged AMOC). We analyse the strength of linear rela-





tionship using a correlation coefficient, $r$, and estimate a significance level for this value using a bootstrapping which assesses
        the probability such a value results if the phases in signals in the $NAM_{10}$ and AMOC are randomly assigned but the overall
        autocorrelation structure is retained. We do this by comparing the $r$-value calculated with real data with those produced from
        a set of synthetic $NAM_{10}$ series. These synthetic data are generated by applying a Fourier transform to the smoothed $NAM_{10}$
        index, randomly shuffling the Fourier phases and subsequently inverse Fourier transforming to generate a surrogate time series

with the same Fourier power spectrum as the real data. Repeating this data generation and calculating the correlation between
        the magnitude of positive extremes in the surrogate $NAM_{10}$s and the 17 year lagged AMOC builds up a PDF for the $r$ value
        which can be used to estimate the significance level for a real $r$ value.

## 3   Results

### 3.1   In-season surface responses to anomalous polar vortex events

We begin by diagnosing the in-season response to anomalous vortex events exhibited by surface variability in the model
        to assess its suitability for studying interactions on longer timescales. Figure 1 shows the mean sea level pressure (MSLP)
        composite differences between strong and weak polar vortex years (figure 1, top row). The composites have been determined
        by selecting MSLP values associated with events in which the daily $NAM_{10}$ values cross the +1.5 (strong) or -2 (weak)
        threshold (see section 2.4). The composite differences demonstrate a significant lagged MSLP response, with strong (weak)

vortex years corresponding to a positive (negative) NAO pattern, in agreement with previous modelling and observational
        studies (see section 1). The NAO anomalies peak in magnitude at a lag of 1-2 months following the vortex anomalies with
        significant anomalies still visible for up to 3 months. There is also a weak positive NAO pattern that leads the vortex anomalies
        by up to one month ($-1-0$ month lags). This may be an indication that an anomalous NAO pattern is a precursor for an
        anomalous vortex, or it could be a response to the initiation of the vortex anomaly since this usually commences in the upper

stratosphere and pre-dates the event's central date defined at 10 hPa. Further exploration of this weak NAO signal is outside the
        scope of this paper. Additionally, a much stronger significant positive anomaly over the Aleutian low (AL) region is evident
        in the month leading up to the vortex anomaly. This signal has been widely studied (Rao et al., 2019) and links the intensity
        of the AL to the strength of vertically propagating planetary waves that subsequently interact with the stratospheric vortex and
        influence its strength. DM21 examined this coupling using the same pi-control simulation as presented here and found a similar

statistically significant relationship between the AL and the frequency of SSWs but the regression coefficients were small in
        comparison with the QBO influence. Here the association between the AL and the vortex strength appears marginally stronger
        (r = 0.39 with the NAM) which may be due to the NAM's ability to capture both types of vortex anomalies (strong and weak).
        In the same study, DM21 found that the AL exhibited minimal decadal to multi-decadal variability that was coherent with the
        decadal to multi-decadal variability of the vortex and for this reason, the role of the AL is not considered in detail in this study.




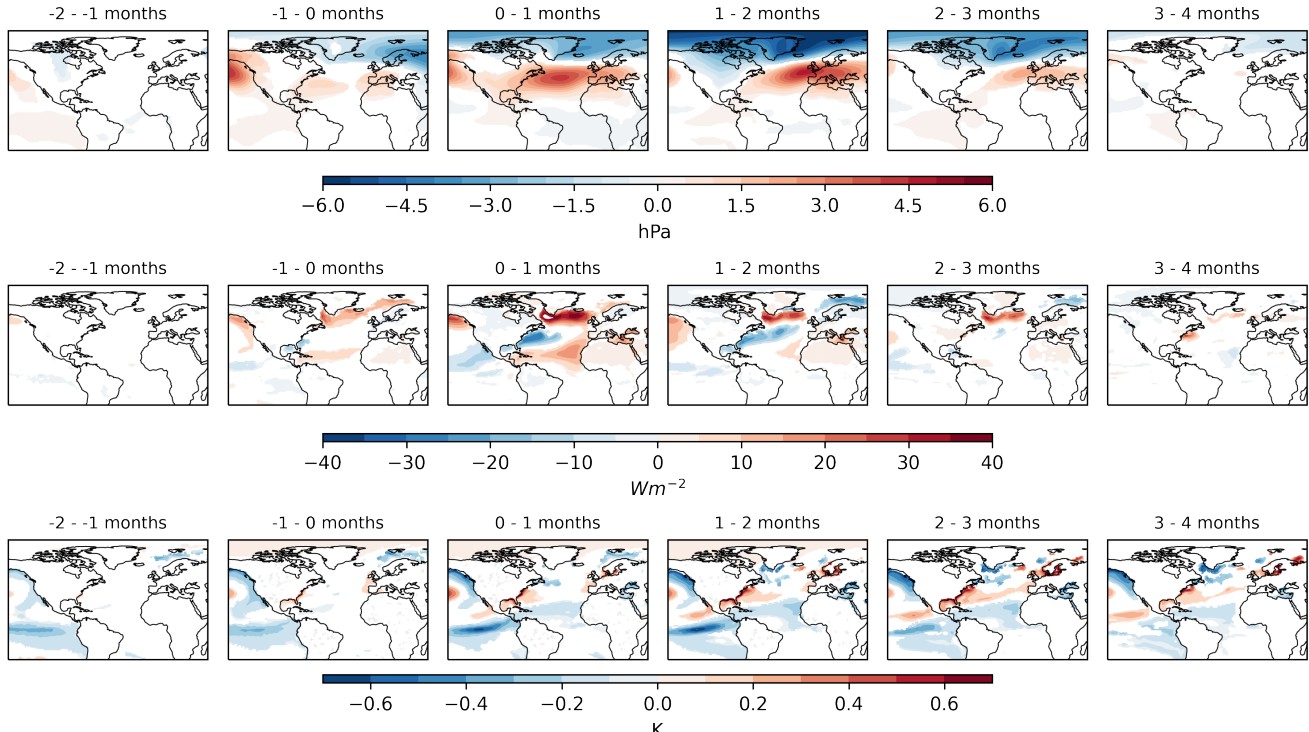

**Figure 1.** Surface patterns associated with anomalous winter stratospheric $NAM_{10}$ events. **Top row**: Monthly mean sea level pressure anomaly (hPa), **middle row**: Ocean-Atmosphere heat flux defined as the sum of latent and sensible heat fluxes ($wm^{-2}$)and **bottom row**: SSTs (K). Coloured shading shows where the composite differences between strong and weak $NAM_{10}$ events are statistically significant at the 95% level under a 2 tailed students t-test. The title of each sub-figure indicates the month range relative to the central date of each $NAM_{10}$ anomaly. Signals at negative times indicate that the surface anomaly leads the stratospheric $NAM_{10}$ anomaly. Signals at positive times indicate that the stratospheric $NAM_{10}$ anomaly leads the surface response.

The model also exhibits significant responses in ocean-atmosphere heat flux (figure 1, middle row). The largest flux anomalies are seen within 30 days (lag 0-1 month) and their spatial pattern resembles that of a North Atlantic tripole with positive anomalies over the subpolar North Atlantic between approximately 50°-65°N, negative anomalies off the east coast of the USA and a second positive anomaly off the coast of North-East Africa. This pattern is consistent with the model response found by Reichler et al. (2012) to anomalous stratospheric $NAM_{10}$ events as well as the pattern associated with positive NAO phases in Delworth and Zeng (2016). As with the MSLP composites, there are visible anomalies 30 days leading up to the identified events (lag -1 - 0 months) both over the North Atlantic and Pacific regions. The Atlantic pattern may correspond to early responses to a disrupted or strengthened vortex as well as possible precursors to events. The Pacific anomalies preceding events are considerably smaller than the Atlantic anomalies and are concentrated over the Aleutian Low region.



The SST response to anomalous stratospheric $NAM_{10}$ events (figure 1, bottom row) over the North Atlantic lags behind the
heat flux anomalies by around 2 months, with the largest amplitude anomalies at around 2-4 month lags. The anomaly pattern
resembles that of the heat flux anomalies (with a change of sign), consistent with a mechanism in which the SSTs respond to the
anomalous heat fluxes. A prominent negative tropical East Pacific anomaly is obvious in the months leading up to anomalous
vortex events together with anomalies that resemble the Pacific Decadal Oscillation (PDO) in the region of the Aleutian Low

(Mantua et al., 1997) and these features persist for several months. Variability in this region is dominated by El Niño Southern
Oscillation (ENSO) variations and a significant body of work (e.g. Domeisen et al. (2019)) has proposed teleconnections
between ENSO and vortex strength, consistent with the type of association exhibited here, i.e. negative (positive) SSTs or la
Niña (el Niño) conditions associated with an anomalously strong (weak) vortex.

## 3.2    Surface impacts of persistent vortex anomalies

The in-season anomaly patterns associated with anomalous stratospheric $NAM_{10}$ events shown in figure 1 confirm that the
model can reproduce the observed influence of vortex anomalies at the surface, particularly over the Atlantic region. We now
extend the analysis to examine decadal-scale variability. Following the approach of Reichler et al. (2012) we smooth the
$NAM_{10}$ index (figure 2) and then select the upper and lower 5 percentiles of this index to identify intervals with a persistent
consecutive strong or weak polar vortex (see section 2.4 for more details). The red and blue dots on figure 2 indicate the

central year of intervals identified with persistent consecutive vortex anomalies (each dot represents the centre of intervals of
approximately 8 years, see section 2.4). Characteristic surface responses associated with these intervals are then analysed by
compiling composites surrounding the central year of each positive and negative interval at lags of -40 (before the intervals)
and 40 (after intervals) years. Calculating the (positive minus negative) composite difference can then be used to assess the
potential surface impacts to observed intervals of persistent consecutive vortex anomalies, such as the consecutive strong

anomalies throughout most of the 1990s and the consecutive weak anomalies in the early 2000s.

        A lead-lag analysis of composite differences in the AMOC strength at three different latitudes is shown in (figure 3). The
figure can be directly compared with figure 4c of Reichler et al. (2012) who suggest that decadal-scale variability in vortex
strength acts to amplify a similar timescale of variability in the AMOC through resonance between the two signals. Similar to

that work, an oscillatory AMOC response to the stratospheric anomalies is evident here, with significant positive anomalies in
the AMOC at 45N and 50N at lags of approximately 3-5 years after persistent $NAM_{10}$ intervals followed by negative anomalies
at lags between 15 and 20 years. This response pattern is more clearly evident by taking the low pass filtered versions of the
AMOC responses (figure 3b, d and f). Even after the high-frequency signals have been removed, there are significant composite
differences of up to 1.5Sv at 15-20 year lags. (We note, however, that this low pass filtering of the AMOC time series reduces

the overall variance so that the threshold for a composite difference to pass the significance test is lower, and this may increase
the responses that are deemed significant shown in figure 3).





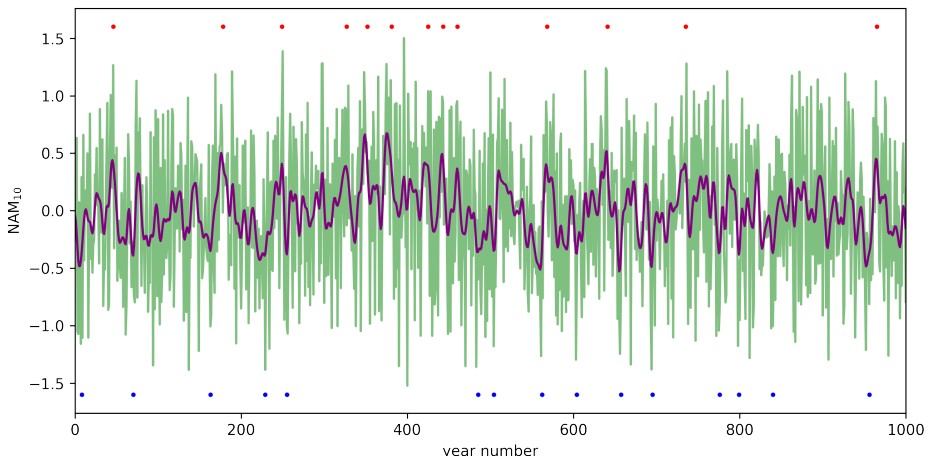

**Figure 2.** Time series of the December-March mean $NAM_{10}$ index (green) level (**green**) and smoothed $NAM_{10}$ (purple) evaluated at the 10hPa. The smoothed series is calculated by applying a Gaussian filter ($\sigma = 2$ years) to the green series. Red and blue dots indicate the occurrence of persistent strong and weak vortex intervals respectively defined as extreme values (top and bottom 5 percentiles) in the filtered $NAM_{10}$ index. interval central years are selected such that at least 10 years lies between consecutive intervals.

Oscillatory response behaviour is not exhibited clearly by the AMOC at 30N, although negative anomalies at lags of 15-20 years after intervals are still visible. Response patterns at this latitude are also significantly smaller than those at 45N and 50N in the filtered composites (figure 3a and b). One possible explanation of this is that the coupling mechanism between the $NAM_{10}$ and AMOC may involve an AMOC response that originates at higher latitudes and then propagates equator-ward which leads to less forcing of the AMOC evaluated further south. Zhang (2010) also note latitudinal differences in the AMOC response so these differences between 30N, 45N and 50N are not unexpected. The AMOC signals at 45N and 50N also exhibit significant positive anomalies preceding the persistent vortex intervals, at a lead of approximately 20 years and this is also present, albeit smaller in magnitude, in the low pass filtered indices. This precursor to persistent $NAM_{10}$ intervals is not found in corresponding results from Reichler et al. (2012) and the role of this feature is considered in more detail in section 3.5.

We now examine this vortex-AMOC teleconnection in closer detail to explore possible physical pathways responsible for an AMOC response to persistent $NAM_{10}$ intervals. Figure 4b shows that there is also an oscillatory response in the NAO. This consists of a positive zero-lag response difference (consistent with figure 1), a significant negative NAO anomaly between lags of 10-18 years followed by a positive NAO response at lags of around 28 years but with smaller amplitude than the zero-lag response.

Also evident from figure 4b is that the oscillatory responses of the NAO and AMOC are similar, with the NAO leading the AMOC response by 2-3 years. Both responses vary with periods of 28-30 years but, interestingly, the negative responses





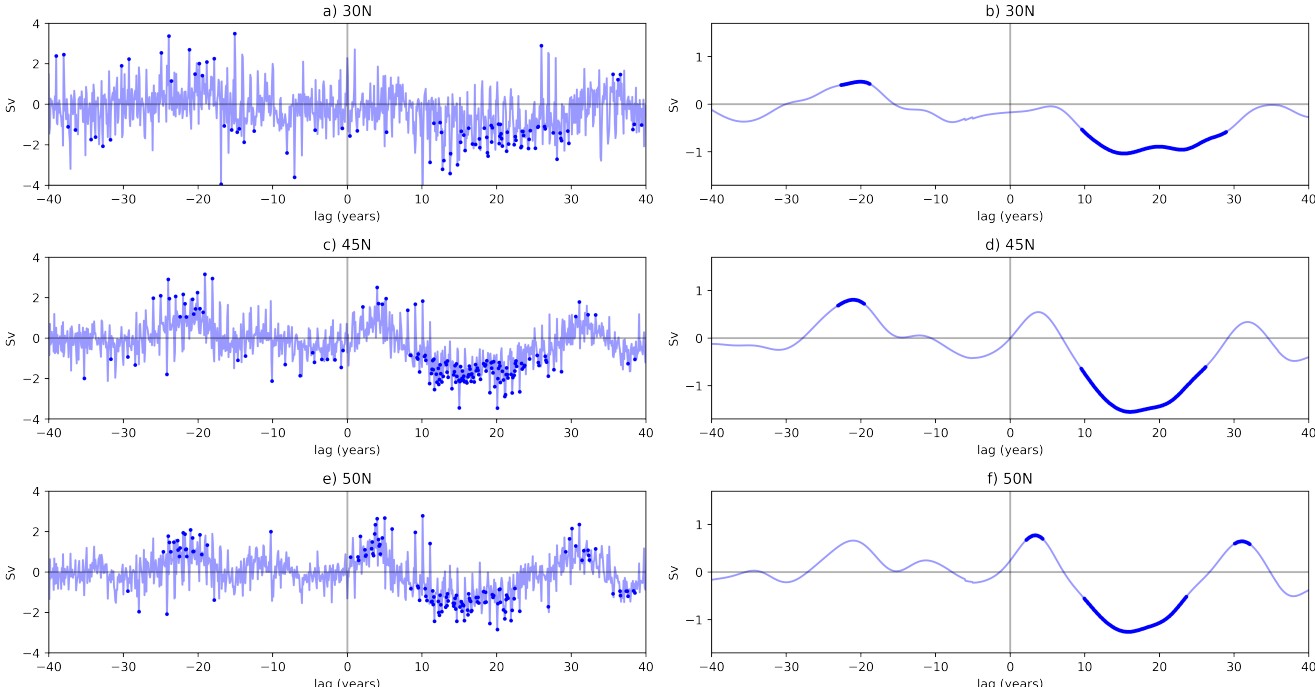

**Figure 3.** Lagged response of the AMOC index to persistent $NAM_{10}$ intervals. Blue time series shows AMOC composite difference values between positive and negative $NAM_{10}$ intervals defined in section 2.4. The $x$-axis denotes the lead (negative values) or lag (positive values) relative to the interval's central year. Black dots denote composite differences significant at the 95% level under a 2-tailed student's t-test. Panels a, c and e show monthly AMOC composites while b,d and f show smoothed AMOC composites using a Gaussian filter ($\sigma = 2$ years).

in both signals are larger and longer-lasting at 10-20 year lags than those at zero and 28-30 year lags. This difference in
the magnitude and persistence suggests that the NAO and AMOC response patterns cannot be explained as a straightforward oscillatory response to the $NAM_{10}$ forcing at zero-lag. If this were the case then the response amplitude would be expected to decay with time and the negative response at 10-20 year lag would be smaller than the initial positive response. Instead, a form of feedback mechanism is required to explain the amplified 10-20 year lagged responses or a resonant mechanism as proposed in Reichler et al. (2012). If such a feedback mechanism were present then one might also expect to see some oscillatory
behaviour in the smoothed $NAM_{10}$ time series, in response to the feedback from the surface. To investigate this, figure 4a shows the corresponding lead-lag difference analysis for the $NAM_{10}$ index (i.e. the smoothed $NAM_{10}$ composited around its own extreme values). This supports the presence of a feedback mechanism since it also exhibits oscillatory behaviour with the same period of around 30 years. However, this oscillation is largely evident from the two positive peaks at zero-lag and 30yr-lag, with the latter being substantially damped. There is no significant response at lags of 10-20 years, where the NAO
and AMOC responses were largest. This suggests that the negative NAO and AMOC responses at these lags are unlikely to be due to resonance with the $NAM_{10}$ signal.





Alternatively, a possible physical pathway could involve an amplifying feedback mechanism between the NAO and AMOC responses. In this scenario, the positive zero-lag NAO response would drive a positive ocean-atmosphere heat flux anomaly over the sub-polar North Atlantic as seen in the response patterns to individual vortex events in figure 1. This heat flux anomaly

would then lead to persistent negative anomalies in the near-surface ocean temperatures as heat is removed from the ocean via variations in wind stress and evaporation. The black line in figure 4 shows the lead-lag difference for ocean-atmosphere heat flux in the region encompassing the Subpolar North Atlantic ($45°$ –$65°$ N, $15°$ –$60°$ W). This region was selected to encompass the region with the largest heat flux response to individual vortex events in figure 1 (see middle row). Figure 4c the corresponding depth profile of ocean temperature response from the same region. A positive heat flux response, as well as an upper ocean

(0-200m depth) cooling, is evident at 0-1yr lags. This heat flux perturbation would, in turn, drive a positive AMOC anomaly at 2-3 year lags via changes in the mixed layer depth and deep convection in the sub-polar North Atlantic, an effect discussed in Delworth et al. (1993) and Medhaug et al. (2012). This increase in AMOC strength would subsequently increase the Labrador Sea temperature via poleward transport of heat. This is confirmed by the positive, deep (down to 2000m) ocean temperature anomaly at a lag of 10-20 years in figure 4c. In turn, the reversal of the Labrador Sea temperatures can feedback onto the NAO

(see e.g. Frankignoul et al. (2013), inducing a negative NAO phase at 10 years lags as the increased Labrador Sea heat content alters the ocean-atmosphere heat fluxes in the same region. Finally, this switch in the NAO phase would lead to a subsequent negative AMOC anomaly via the same heat flux mechanism outlined above for an opposite NAO phase. This sequence of feedbacks would thus act to enhance the persistence and magnitude of the secondary extreme in the NAO and AMOC. Reichler et al. (2012) also briefly suggest a similar mechanism to account for the AMOC response in their simulations, but involving a

negative feedback of the AMOC onto itself as well as a role for the NAO.

### 3.3 Response to strong and weak vortex intervals

So far, we have considered composite difference responses to persistent strong and weak vortex intervals. However, we know that the vortex evolution during strong and weak vortex years is very different and this is likely to lead to differing interactions

with surface and ocean variability. The surface responses to the two extremes are therefore unlikely to be equal and opposite. For example, weak vortex winters are mostly associated with SSWs whose impact at the surface is observed on average 0-60 days after their central date (Baldwin and Dunkerton, 2001). Furthermore, the vortex often exhibits a pre-conditioned state (see e.g. Charlton and Polvani (2007) and Bancalá et al. (2012)) in which it becomes anomalously strong in the weeks running up to an SSW. So the timing of SSW events within a given season will dictate both the overall strength of the $NAM_{10}$ measured over

the winter season (which we use to construct the persistent $NAM_{10}$ index) and the overall strength of the subsequent surface response. In contrast, winters that exhibit an anomalously strong vortex will, on average, exhibit such behaviour throughout the whole season so the impact on the surface will be present for a larger fraction of the winter season.

To assess the influence from each type of vortex extreme separately, figure 5 shows the lead-lag composite analysis of the NAO, AMOC, $NAM_{10}$ and heat flux signals for the persistent positive (strong vortex) and persistent negative (weak vortex)

$NAM_{10}$ intervals separately. The AMOC patterns associated with each $NAM_{10}$ type are slightly different. The persistently



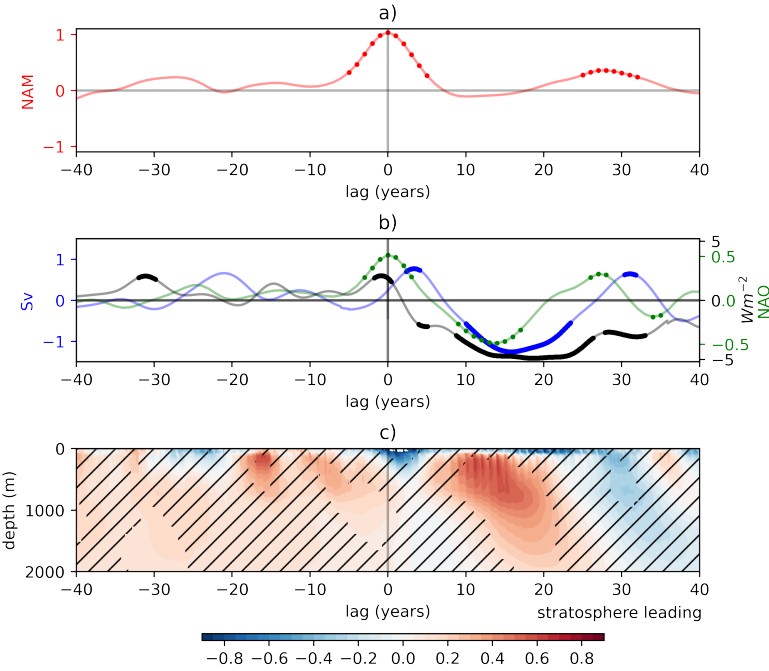

**Figure 4. a**: Composite differences of the smoothed $NAM_{10}$ index around extreme $NAM_{10}$ intervals (positive - negative intervals). **b**: Like figure 3f for the AMOC at 50N (blue), the Dec-Mar mean NAO index and the ocean-atmosphere heat flux (sum of latent and sensible fluxes) averaged over an Atlantic box defined by 45°–65°N, 15°–60°W. All indices are smoothed with a Gaussian filter ($\sigma$ = 2 years). **c**: lagged responses of ocean temperature anomaly depth profiles to persistent $NAM_{10}$ intervals. Composite differences between strong and weak intervals are shown for the same Atlantic box as the heat flux index. Hatching indicates composite differences that are not significant at the 95% level under a 2-tailed student's t-test.

strong vortex composite shows clear oscillatory behaviour with a period of approximately 28 years. These patterns resemble many of the features observed in the AMOC composite differences (figure 4d). Specifically, a positive AMOC anomaly is present at lags of 2-3 years, a negative AMOC anomaly at 15-20 years and a second positive anomaly at approximately 30 years. On the other hand, the persistently weak vortex composites exhibit a more complicated response, with double-peaked minima at lags of -20 and -11 years and double-peaked maxima at 14 and 25 years. The weak vortex composites also exhibit no significant AMOC response at 2-3 year lags, unlike the strong vortex composites. Both event types are associated with NAO anomalies at zero-lag but the response to strong vortex intervals is larger in magnitude, which is consistent with the larger 2-3 year lagged AMOC response to this event type. The 0-lag NAO response is followed by an extreme of the opposite sign at approximately 16yr and 14yr lags for strong and weak intervals respectively. As with the AMOC composites, the NAO response to persistently strong vortex intervals exhibits a pronounced oscillatory behaviour of periods around 28 years. The corresponding $NAM_{10}$ analysis also shows oscillatory behaviour with periods of around 28 years. (We note that the $NAM_{10}$ results show statistical significance at both lead and lag times. However, the lead/lag interpretation is less meaningful in this





case since the $NAM_{10}$ is used both as the signal and in the selection of the composites. The significance at both lead and lag times simply confirms that there is oscillatory behaviour). The double-peaked behaviour of the AMOC associated with

weak intervals is also reflected somewhat in the sub-polar North Atlantic heat flux response (figure 5c) with positive response peaks at approximately 11 and 20-year lags. There is also a 0-lag heat flux anomaly associated with weak vortex intervals corresponding to the negative NAO response but the corresponding response to strong intervals is not significant.

The asymmetry between the AMOC and NAO responses to persistently strong and persistently weak vortex intervals and the complexity of the separate responses show that the interactions between the $NAM_{10}$ and these surface modes are complex,

with some suggestion of oscillatory behaviour on different timescales. In the following sections, we address these complexities in more detail by analysing the frequency spectra of the time series and show that some of these complexities can be explained in terms of the non-stationarity of the signals.

### 3.4 Non-Stationary Variability

In an analysis of this same UKESM pi-control simulation, DM21 showed that variability of the stratospheric polar vortex occurs on a range of timescales and is highly non-stationary. Although the composite analysis presented above shows oscillatory behaviour with periods of approximately 30-yr the results are likely complicated by the presence of non-stationary variability at other periodicities. We therefore analyse the frequency characteristics of the filtered $NAM_{10}$ index. Figure 6 shows the wavelet power spectrum of this index. It reveals variability on a range of timescales. As expected from the composite analyses,

the spectrum exhibits intermittent power throughout the whole simulation at periods between 15 and 40 years. There is also significant power corresponding to a period of approximately 90-100 years that persists for ∼300 years of the simulation (year numbers 520-820; approximately 3 cycles) as well as power at the 50-year timescale that persists for 120 years (year numbers 500-620; approximately 2 cycles).

We note that much of the ∼30-yr periodicity seen in the strong composite analysis is likely associated with significant wavelet

power in the interval between years 300 and 410 because this interval displays the largest number of positive extremes in the smoothed $NAM_{10}$ time series (compare the 30-yr wavelet power in figure 6 with the red dots in figure 1). The large number of elevated $NAM_{10}$ extremes selected in this interval is also likely affected by the presence of extremely long timescales variability. Qualitative inspection of figure 2 shows that the underlying $NAM_{10}$ amplitude increases from around year 200, reaches a peak at ∼year 380 and thereafter declines. This means that years within this interval are more likely to reach the top

5 percentile and qualify as an anomalously strong vortex. The origin of this multi-centennial variability is unclear and robust analysis of such a low frequency signal is difficult as the wavelet power of this multi-century variability is located mostly outside the so-called "cone of influence" that marks the boundary at which edge effects become significant.

We also analyse the frequency characteristics of AMOC variability. Figure 7 shows the corresponding wavelet spectra of

the AMOC at 50N and also the cross wavelet spectra between the filtered $NAM_{10}$ and the AMOC which gives a time-varying





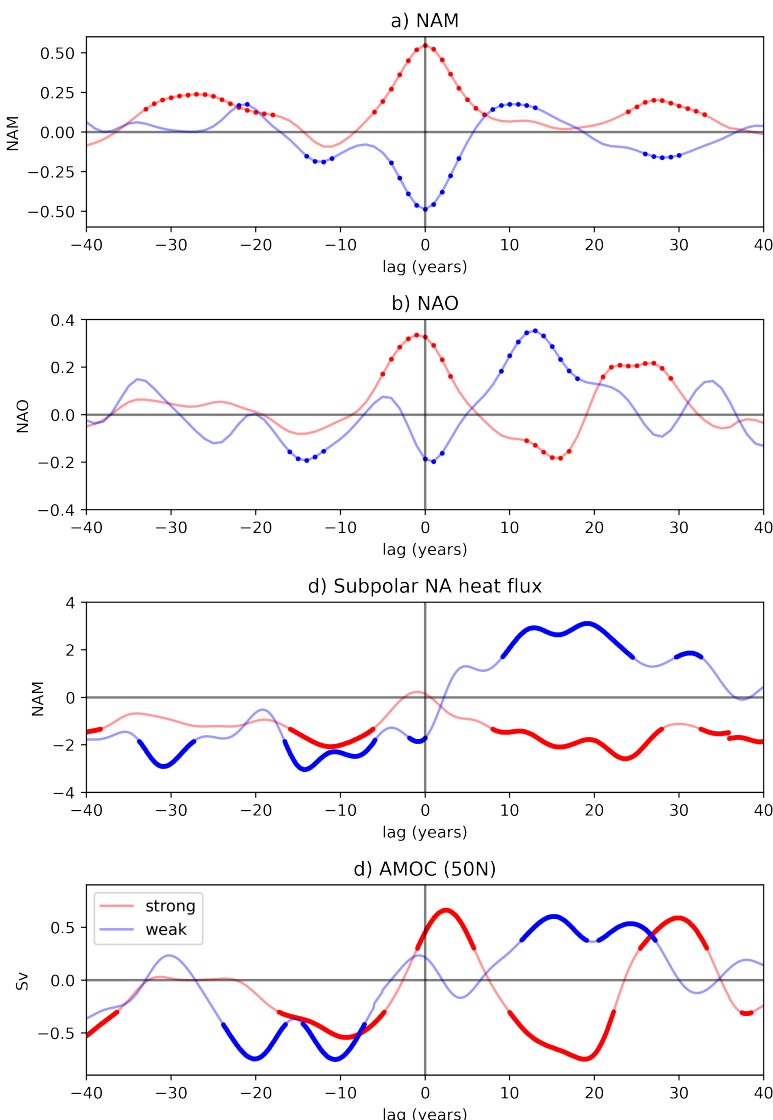

**Figure 5. a**: Composites of (a) Gaussian smoothed AMOC anomalies at 50N, (b) NAO, (c) $NAM_{10}$ and (d) subpolar NA heat flux associated with persistent vortex intervals of different types. On each sub-figure, the red (blue) plots show the lead/lag responses to composites of strong (weak) persistent $NAM_{10}$ intervals. Solid dots denote composite anomalies are significant to the 95% level under a 2 tailed student's t-test.

measure of co-variability between the two indices at different periods. The wavelet spectrum for the AMOC at 50N exhibits a peak in spectral power corresponding to approximately 130 years that persists for nearly 400 years of the simulation (and also at longer periods up to 250 years but boundary effects are an issue at these multi-centennial timescales, as discussed above). There are also portions of significant power at approximately 30-year and 50-year periods, both of which persist for

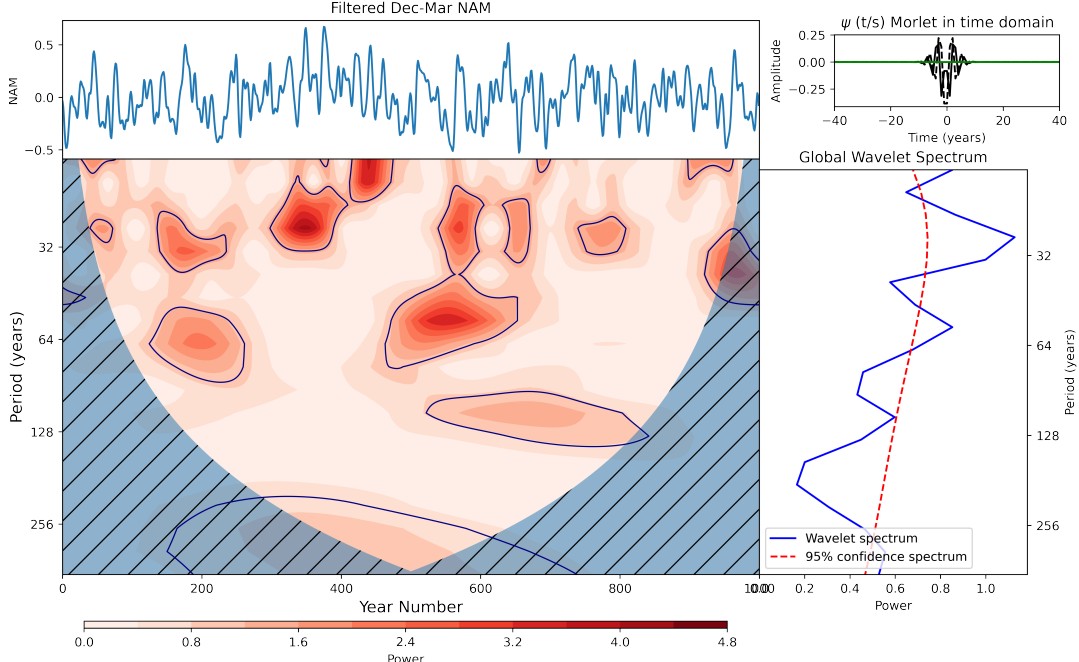

**Figure 6. Top left**: Dec-Mar $NAM_{10}$ values smoothed with a Gaussian filter ($\sigma = 2$ years). **Bottom left**: Wavelet power spectrum of time series in the top left sub-figure. Hatching represents areas outside the cone of influence in which edge effects are significant and power should not be considered. blue contours represent the 95% confidence level assuming mean background AR1 red noise. **Top Right**: Morlet wavelet used for the wavelet transform in the time domain. **Bottom right:** Global power spectrum, the wavelet power averaged over the whole simulation (blue line), and global 95% confidence spectrum (red dashed line).

approximately 2 cycles (∼60 years and ∼100 years respectively) which are also apparent in the global power spectrum. We also note that the main 30-yr power comes from the 300-400 year interval, coinciding with the interval of most activity in both the $NAM_{10}$ analysis (figure 6) and the selected high $NAM_{10}$ percentiles (figure 1).

The cross power spectrum between the filtered $NAM_{10}$ and the AMOC shows 3 distinct features corresponding broadly to the three timescales prominent in the individual spectra of both indices. Significant cross power is evident at 90-100 year periods for approximately 350 years (between 450 and 800 years; 3-4 cycles). The phase relationship between the signals (indicated by arrows on figure 7) within this portion of the cross spectrum show a mixture of left-pointing arrows in the earlier portion, which indicate an anti-correlated relationship ($\pi$ out of phase) and downward pointing arrows in the later portion that indicate a $\frac{\pi}{2}$ phase relationship. This later phase relationship can be interpreted in a number ways, with maxima in the AMOC leading to maxima in the $NAM_{10}$, minima in the $NAM_{10}$ leading to maxima in the AMOC or maxima in the $NAM_{10}$ index coinciding with maxima in the rate of change of the AMOC (see section 2.2 for more details of the cross spectra arrows and how they are derived). There is also significant cross-spectral power centred around 30 years (between 300-400 years; 3 cycles)





and 50 years (between 500-600 years; 2 cycles). In contrast to the 90-yr periodicity, the phase arrows point to the right and slightly upwards, indicating that maxima in the $NAM_{10}$ index lead maxima in the AMOC by a small fraction of the cycle. This phase relationship is consistent with the composite analysis presented in figures 3 and 5 that indicated that a positive (negative)
$NAM_{10}$ leads to a positive (negative) AMOC response approximately 2-3 years later. The wavelet spectra for the AMOC at 30N and 45N are provided in figures A1a and A2a. They show broadly the same features as that of the AMOC at 50N however it is notable that the AMOC at 30N does not exhibit significant variability on the 50 and 30-year timescales. This is reflected in the cross spectrum with the smoothed $NAM_{10}$ index which shows minimal cross power on these timescales (figure A1b).

To understand these non-stationary signals in the context of the proposed mechanism for vortex-AMOC interactions involv-
ing the NAO, we also analyse the power spectrum for the Dec-Mar NAO (figure 8). The wavelet power spectrum for the NAO (figure 8a) exhibits a portion of significant power at periods of 30 years between ∼300-400 years as well as a feature at 50 years between ∼500-600 years similar to the $NAM_{10}$ and the AMOC. Furthermore, the cross power spectrum between the NAO and the $NAM_{10}$ indicate that signals in the two indices on the ∼30 and ∼50 year periods are also coincident in time for the 70-100 years they persist for. The phase relationship between these signals is small (arrows pointing to the right) indicating
an in-phase relationship between the indices, which is consistent with the zero-lag relationship between the NAO and filtered $NAM_{10}$ extremes presented in the composite analysis (figures 4 and 5). The NAO wavelet analysis shows no significant power on the 90-100 year timescale. This suggests that the co-variability between the $NAM_{10}$ and AMOC on these longer timescales does not involve the NAO and is likely to arise through different mechanisms. We return to examine this feature in more detail in section 3.5.

Results from this wavelet analysis may also explain the different behaviour of the AMOC around persistent positive and negative $NAM_{10}$ intervals (figure 5). Firstly, we note that the contribution to the composites from the prime interval exhibiting ∼30-year oscillatory behaviour (∼300-400 years) comes solely from a collection of persistent strong $NAM_{10}$ intervals (see the red dots on figure 2). In contrast, a high proportion of the weak $NAM_{10}$ contributions to the composite analysis (8 out of 13) occur within the interval between ∼500-600 years which exhibits variability at both 50yr and 90yr periodicity. The
complicated double-peaked behaviour of the AMOC response following persistent weak vortex intervals can now be better understood: The double minima in AMOC response at 10yr and 20yr leads and at 15yr and 25yr lags can now be explained as manifestations of the 50-yr and 90-yr AMOC responses e.g. a half-cycle between the minimum at 10yr lead and maximum at 15yr lag (a half-cycle of 25 years) gives a periodicity of 50yrs, while a half-cycle between the minimum at 20yr lead and 25yr lag (thus a half-cycle of 45yrs) give a periodicity of 90yrs. Additional support for this interpretation comes from the fact that
the NAO composite analysis (figure 5b) shows a response that corresponds broadly with the AMOC minimum at 10yr lead and maximum at 15yr lag, suggesting a mechanism that involves the NAO on the 50-yr timescale but there is no corresponding response in the NAO at the 90-yr periodicity, in agreement with the wavelet spectra and cross spectra in figure 8 which indicates that the NAO does not exhibit variability on timescales of ∼90 years.




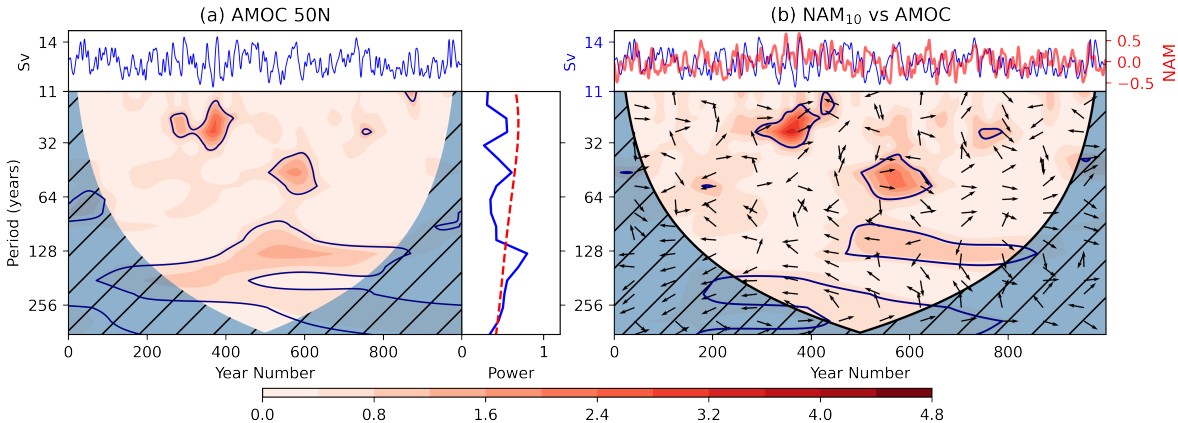

**Figure 7. (a, top)**: AMOC time series at 50N, **a, bottom left**: Wavelet power spectrum (shaded contours represent wavelet power and yellow contours the 95% significance level compared to an AR1 process), **a, bottom right**: global wavelet power spectrum (blue) and 95% confidence level (dashed red). **b**: Cross spectra between Filtered Dec-Mar $NAM_{10}$ series and the AMOC index. **b, top**: $NAM_{10}$ and AMOC time series. **b, bottom**: Cross power spectrum. Shading indicates cross power, yellow contours the 95% confidence interval and arrows the relative phase angle between signals in the time series (to the right: in phase, vertically upwards: $\frac{\pi}{2}$ out of phase with positive peaks in the $NAM_{10}$ leading those in the AMOC, to the left: $\pi$ out of phase, vertically downwards: $\frac{\pi}{2}$ out of phase with positive peaks in the AMOC leading those in the NAM).

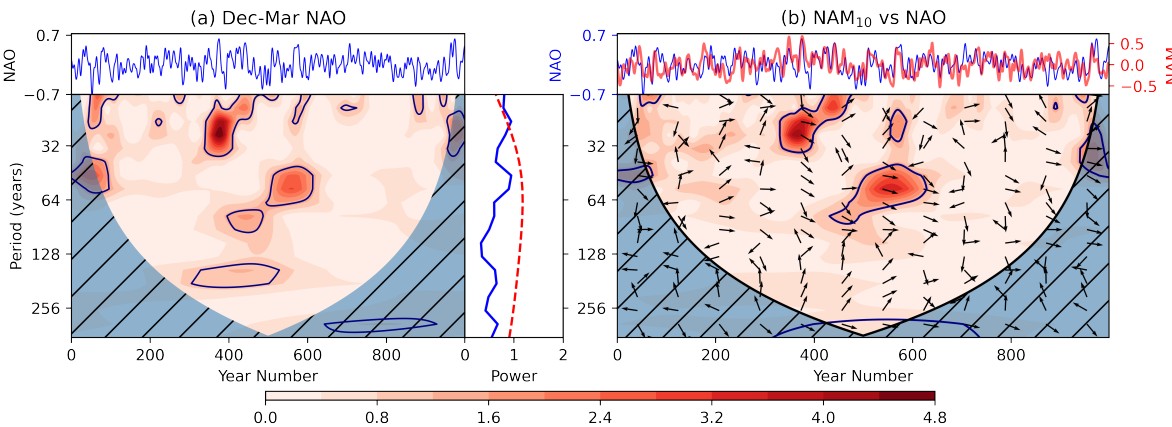

**Figure 8.** like figure 7 for the Dec-Mar NAO index. (a) shows the wavelet power spectrum of the NAO and (b) the cross power spectrum between the NAO and the $NAM_{10}$ index.





## 3.5 Surface Forcing of the Stratosphere

The absence of an NAO signal that corresponds to the 90yr periodicities seen in both the $NAM_{10}$ and AMOC wavelet spectra and in the AMOC-NAM cross spectrum (figure 7) indicates that the mechanism for the stratosphere - AMOC teleconnection on this longer timescale may be distinct in nature to those observed at the 30 and 50-year periodicities. As noted earlier, the

phase relationship associated with this feature is also different to the shorter timescales ($\frac{\pi}{2}$ out of phase) suggesting that the direction of causality is also switched i.e. that the AMOC leads the stratosphere response on these timescales rather than vice versa. To study this more closely we investigate possible pathways involving variability on this timescale.

In an earlier analysis of this UKESM pi-control simulation DM21 highlighted the ∼90yr variability in the frequency of SSWs and demonstrated that this was closely associated with similar timescale variations in the amplitude of the QBO (and in

particular, the westerly QBO phase). Figure 9a shows the wavelet spectrum of the same QBO index employed by DM21 after smoothing with the same Gaussian filter utilised throughout this work (see DM21 figure 12 and also section 2 for a description of how the QBO index was derived). A portion of significant power at ∼90-year periods persists for approximately 200 years of the simulation between years 600-800, which coincides with the significant response in the same interval of the $NAM_{10}$ and AMOC spectra at 90yr periodicity. Cross spectra of the QBO index with the smoothed $NAM_{10}$ index (figure 9b) also

corroborates the findings of DM21, with coincident signals at the 90-year timescales and right-pointing arrows that indicate an in-phase relationship, so that an interval of persistently strong positive (westerly) QBO anomalies coincides with an interval of persistent positive (strong) polar vortex anomaly. The sign of this teleconnection is consistent with the well-known Holton-Tan teleconnection (Lu et al., 2008, 2014) but it is present on much longer timescales. DM21 showed that this long-term variability originates primarily from long-term variations in the strength of the westerly QBO phase.

While the long-timescale QBO-vortex teleconnection was demonstrated by DM21 (and confirmed here), the cause of the long-term westerly QBO variability was not established. In a preliminary investigation, DM21 performed a wavelet analysis of equatorial SSTs in various regions, including the equatorial East Pacific, to explore whether the ∼90-yr QBO variability could be explained by SST triggering of convective activity that generates the gravity and other equatorial waves that contribute to the QBO. However, no 90-year periodicity in equatorial SSTs was found. DM21 also performed a corresponding analysis of an

index representing the strength of the Aleutian Low, to explore whether that could explain the 90-yr signals in both the QBO and the polar vortex via a modulation of the strength of large-scale planetary wave forcing, but no periodicity at 90 years was found in the AL index. Nevertheless, the extremely long period of the QBO and vortex variations suggests a driving mechanism that is likely linked to the ocean, because of the characteristic long oceanic timescales. We therefore extend the investigation of DM21, to pin down the mechanism that links the observed variability on 90-yr timescales in the AMOC, the QBO and the

polar vortex.

To extend this investigation we examine the variability of the East Pacific top-of-atmosphere outgoing longwave radiation (OLR) as a proxy for deep convection (instead of using the East Pacific SSTs, as in DM21). When deep convection is enhanced, cloud top height is increased and therefore OLR is reduced. Figure 9c shows the wavelet analysis of the Sep-Nov OLR in the East Pacific. It exhibits 90-year periodicity, as well as significant cross power with the smoothed QBO amplitude and the





NAM$_{10}$ index (figures 9d,e). The signals in the OLR and both the QBO and NAM$_{10}$ are anti-correlated (left-pointing arrows indicating a $\pi$ phase difference). This is consistent with reduced OLR (increased deep convection) leading to greater QBO amplitude through increased wave forcing. The corresponding cross spectra between the AMOC and the OLR metric (figure 9f) also indicates a significant portion of cross power in the interval $\sim$600-800 years co-located with the feature seen in the NAM$_{10}$ spectrum (see the dashed contours in figure 9 which indicate the region with significant power in the NAM$_{10}$ spectrum). The phase relationship, in this case, is mostly $\frac{\pi}{2}$ (the majority of arrows pointing upwards) indicating that one of the quantities depends on the time rate of change of the other. This result is similar to the study of Timmermann et al. (2005) who found a sensitivity of the equatorial Pacific region to periodic forcing of the AMOC. Their study showed a dependence of the Pacific thermocline on the rate of change of the AMOC. (We note, however, that the $\sim$7.5 Sv AMOC perturbation imposed in their study was considerably larger than the AMOC variations in our simulation). Similarly, a lagged, cross-basin connection between the NA overturning circulation and Pacific Sea Surface Height (SSH) was proposed by Cessi et al. (2004) who interpreted it in terms of the propagation of oceanic Kelvin and Rossby waves with anomalies communicated between Atlantic and Pacific via the Indian Ocean as well as through the Drake Passage. We therefore suggest this as a possible pathway for the influence of the AMOC on the polar vortex at 90-yr timescales in this simulation, via modulation of deep convection in the East Pacific that influences the amplitude of the QBO. A more detailed examination of the intermediate steps in this proposed physical pathway is required to confirm this, but this is outside the scope of the study.



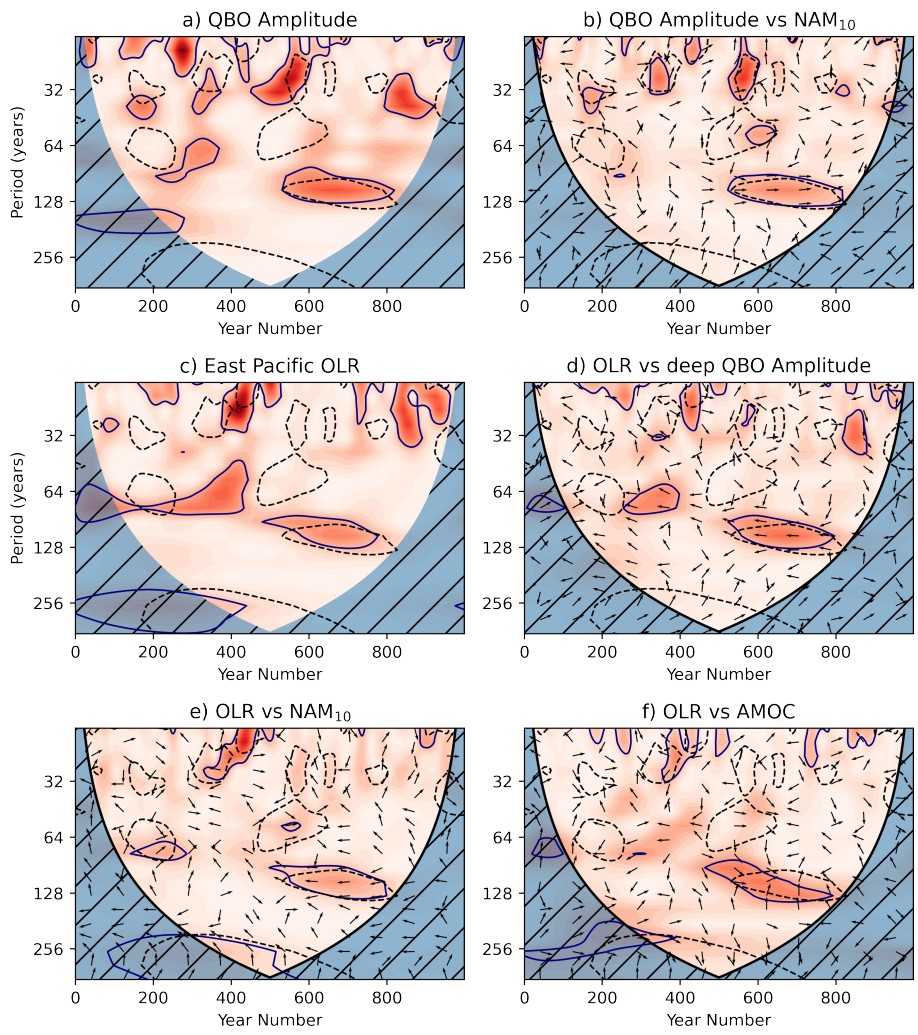

**Figure 9.** (a): Wavelet power spectrum for the Sep-Nov deep QBO amplitude (see section 2.4). (b) cross power spectra between deep QBO Amplitude (Sep-Nov) and smoothed NAM$_{10}$ (Dec-Mar). (c): Wavelet power spectrum for the Sep-Nov Area weighted average equatorial East Pacific OLR (see section 2.4). **d-f**: cross power spectra between combinations of East Pacific OLR, deep QBO Amplitude (both Sep-Nov means), AMOC at 50N (annual mean, all months) and NAM$_{10}$ (Dec-Mar) indices all smoothed with the same Gaussian filter as is used throughout ($\sigma = 2years$). Indices involved are indicated by the sub-figure titles. Shading indicates the cross power, using the same colour scale as in figures 6-8. Solid contours indicate the 95% confidence interval for the power spectrum and dashed contours show the 95% confidence interval for the NAM$_{10}$ spectrum, for ease of comparison. Arrows indicate the relative phase angle between the signals in the indices.



## 3.6 Contribution of the stratosphere to recent AMOC changes

Our analysis of the UKESM simulation has identified co-variability between modes of variability in stratospheric circulation and the AMOC. Intervals in which the winter stratospheric polar vortex is consistently strong are, on average, followed by an
extended negative anomaly in AMOC strength with a lag of approximately 15-20 years (figure 3f). Recent observations of the AMOC have shown a negative trend in circulation strength of approximately $-2.7Sv$ between 2004 and 2012 (Smeed et al., 2018) before a marginal recovery after 2012 (Smeed et al., 2019). Modelling studies have proposed a key role for anthropogenic forcing in AMOC slowdown over the 20th century and into the future (Liu et al., 2017; Bakker et al., 2016; Liu et al., 2019). However, the drivers of observed AMOC trends in the 21st century are not well understood. The results shown in the previous
sections suggest the possibility that stratospheric variability has also contributed to the observed AMOC changes, in response to nearly a decade of strong vortex years in the 1990s followed by a sequence of years with a weak, disturbed vortex in the early 2000s.

Using the relationship seen in the model between the modelled NAM, NAO and AMOC we can now compare the observed NAM, AMOC and NAO indices for the interval 1979-2020 from the ERA5 and RAPID array datasets and assess the potential
contribution of stratospheric variability to the observed AMOC trend (figure 10). The $NAM_{10}$ index (red bars) is characterised by an interval of strong vortex winters between 1988 and 1997 in which all but 2 winters exhibited a positive $NAM_{10}$. This interval also contains no SSWs (Pawson and Naujokat, 1999). This is followed by a run of winters between 1998 and 2005 which exhibit anomalously weak $NAM_{10}$ values with SSWs almost every year Manney et al. (2005). The filtered $NAM_{10}$ index (red dashed line) reflects the presence of these time intervals with a peak in positive values centred around 1995 followed
by a negative extreme centred around 2003. The smoothed NAO (green dashed line) from the same dataset reflects some of these variations in the $NAM_{10}$ with positive NAO extremes in the 1990s. The long-term envelope of the NAO does not remain positive for as long as the NAM, primarily due to the anomalous negative NAO in 1996 (Halpert and Bell, 1997). The presence of this anomalous negative NAO in 1996 and the absence of a clear NAO anomaly in the following year despite the presence of strong positive anomalies in the stratospheric $NAM_{10}$ is indicative of the range of other factors that influence the NAO in
addition to the vortex. The AMOC strength estimated from the Rapid Array observations between 2005 and 2019 is also shown (blue curve) and shows a negative trend between 2005 and 2012 followed by a recovery from 2012 onwards (Smeed et al., 2018, 2019).

The interval of observed consecutive strong NAM winters in the 1990s is anomalous in the reanalysis period although the
available data record is rather short to allow a robust assessment. The amplitude and longevity of the observed anomaly are also large when compared to the UKESM simulation. Only 2 intervals in the UKESM simulation exhibit at least as many consecutive winters with strong (high $NAM_{10}$) conditions. These 2 intervals occur in the 300-400 year interval (centred around years 349 and 376) as shown in figure 11a. They each exhibit a sequence of 10 consecutive Dec-Mar anomalously positive $NAM_{10}$ values. The 2nd interval exhibits 14 strong or marginally weak consecutive winters with the allowance for the small negative $NAM_{10}$
value at year number 379. The presence of these two intervals is reflected in the smoothed $NAM_{10}$ values (figure 11b) and they





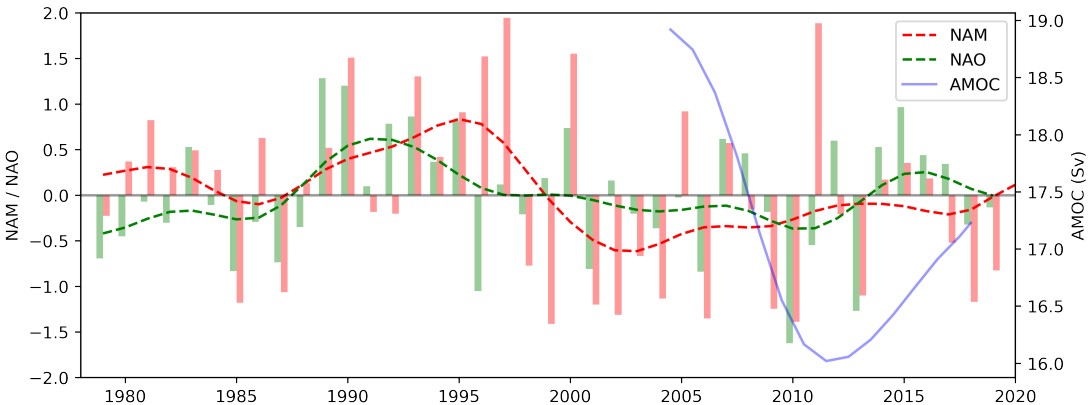

**Figure 10.** Time series of the Dec-Mar NAO index (green bars), $NAM_{10}$ index (red bars) from the ERA5 dataset. Dashed lines correspond to indices shown by bars smoothed with a Gaussian filter ($\sigma = 2$ years). Also included is the annual AMOC time series at 26N from the rapid array dataset (blue).

represent the two of the three largest values of the filtered $NAM_{10}$ index (0.66 and 0.67). The corresponding smoothed AMOC index during these 2 intervals (blue curve in figure 11a) shows a positive AMOC response at lags of 2-3 years followed by a negative response at 17-20 years, in good agreement with figure 3f. The negative responses at 17-20 years following these 2 intervals of strong $NAM_{10}$ years are the 1st and 3rd greatest in magnitude compared to all other responses to persistent strong intervals. This is confirmed by figure 11c which shows the lagged AMOC response following all of the identified intervals with persistent positive $NAM_{10}$ anomalies (the two identified around 349 and 376 years are shown in black).

To estimate the response amplitude of the AMOC to an interval of persistently strong vortex winters, figure 11d shows a scatter plot of the central $NAM_{10}$ index of these intervals against the AMOC anomaly at 50N lagged by 17 years. This reveals a strong linear relationship ($r = -0.908$) between the size of the persistent vortex anomaly and the subsequent negative anomaly in the AMOC at 50N 17 years later. The PDF of surrogate correlations used to assess the statistical significance (following the method outlined in section 2) is displayed in figure A3a along with the correlation generated by the observed $NAM_{10}$ data. It shows that the $r$ value lies well outside the distribution of surrogate correlations, indicating the high level of significance of the linear relationship.

A linear regression analysis on this data yields an estimated relationship between the variables which satisfies

$$AMOC'_{+17} = -6.54NAMmax + 3.11, \tag{12}$$

where $AMOC'_{+17}$ is the 17 year lagged AMOC anomaly at 50N and $NAMmax$ is the magnitude of the positive extreme in smoothed $NAM_{10}$ at the centre of each interval. We can then use this relationship to predict the AMOC response to the observed sequence of strong vortex years in the 1990s. The maximum smoothed $NAM_{10}$ occurs in 1996 so, using this relationship, the





maximum AMOC response associated with the stratosphere would be expected 17 years later (2013) with an amplitude of

-0.89 Sv (figure 11d, purple point).

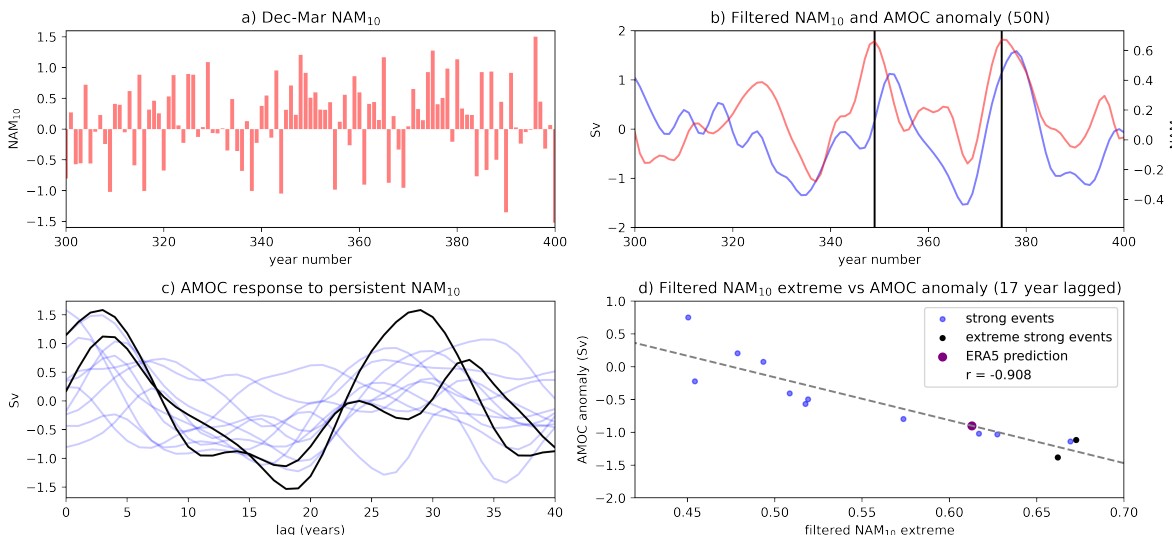

**Figure 11. a**: Dec-Mar $NAM_{10}$ index (red bars) from the UKESM simulation between year numbers 300 and 400. **b**: AMOC (blue) and $NAM_{10}$ (red) indices smoothed with the Gaussian filter ($\sigma = 2$ years) between year numbers 300 and 400 of the UKESM simulation. Black vertical lines show the location of the largest smoothed $NAM_{10}$ **c**: AMOC response to persistent strong $NAM_{10}$ intervals. Light blue lines denote lagged AMOC responses of the AMOC from the whole UKESM simulation, black lines show AMOC responses to $NAM_{10}$ intervals marked in **b** by vertical black lines. **d**: Scatter plot of filtered $NAM_{10}$ index values occurring at persistent strong $NAM_{10}$ intervals throughout the whole UKESM simulation ($y$-axis) against AMOC anomalies at 50N lagged 17 years after persistent intervals' central year ($x$-axis). Blue points indicate persistent intervals and black dots represent the 2 intervals displayed in b. The dotted line represents the linear line of best fit for the points in black and blue. Also included is the 17-year lag AMOC anomaly predicted by projecting the regression coefficients used to construct the linear fit onto the maximum smoothed $NAM_{10}$ index in the ERA5 dataset (purple point).

This prediction suggests that approximately 30% of the observed reduction in AMOC strength between 2005 and 2013 (0.89 Sv compared with 2.9 Sv in total) could be due to the response of the ocean to persistent forcing from consecutive strong vortex winters that occurred during the 1990s. We note, however, that our derivation of the modelled vortex-AMOC relationship is

based on the AMOC response at 50N, where the response amplitude is largest, whereas the Rapid Array dataset provides a measurement of AMOC strength at 26N. Figure 12 shows the scatter plot of filtered $NAM_{10}$ extreme magnitudes and lagged AMOC responses from the model at 30N. At this latitude, the linear relationship is significantly weaker than at 50N ($r = 0.652$ for 30N vs $r = 0.908$ for 50N) but the correlation coefficient remains significant at the 95% level. The predicted contribution from the strong vortex interval in the 1990s to the AMOC strength at 30N is reduced to $-0.49Sv$, (figure 12, purple dot)

suggesting that approximately 17% of the negative trend in the RAPID AMOC data may be due to stratospheric forcing from the 1990s. This is consistent with the composite analysis in figures in 3a and b which indicate that the modulation of the AMOC by the smoothed $NAM_{10}$ is less pronounced at 30N than at higher latitudes.

A similar analysis of the relationship between the magnitude of smoothed negative (weak) $NAM_{10}$ extremes and the lagged AMOC response (not shown) yields a much weaker relationship ($r = -0.21$) that is not statistically significant. This asymmetry
in the vortex-AMOC relationship between extreme positive and negative NAMs is perhaps not surprising, given that the surface impact of SSWs (that give rise to the negative NAM events) depends on the timing of the SSW within the winter season whereas strong positive NAM intervals exhibit strong vortex conditions throughout the winter.

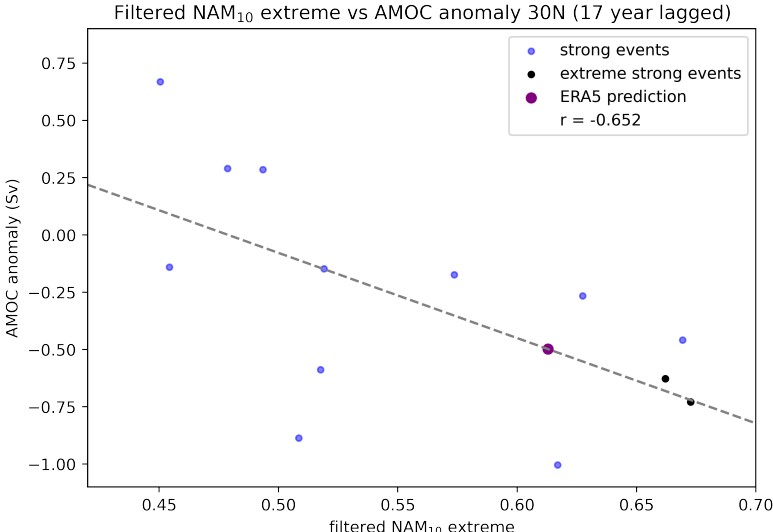

**Figure 12.** Like figure 11d for AMOC responses at 30N: Scatter plot of filtered $NAM_{10}$ index values occurring at persistent strong $NAM_{10}$ intervals throughout the whole UKESM simulation ($y$-axis) against AMOC anomalies at 30N lagged 17 years after persistent intervals' central year ($x$-axis). Blue points indicate persistent intervals and black dots represent the 2 intervals displayed in b. The dotted line represents the linear line of best fit for the points in black and blue. Also included is the 17-year lag AMOC anomaly predicted by projecting the regression coefficients used to construct the linear fit onto the maximum smoothed $NAM_{10}$ index in the ERA5 dataset (purple point).

## 4    Summary and Conclusion

In this study we have analysed the influence of persistent polar vortex extremes on surface and ocean circulation in a 1000-year pi-control simulation of UKESM1. Persistent vortex anomalies are identified using a smoothed $NAM_{10}$ index which characterises intervals of approximately 6-8 years during which the NH winter vortex is anomalously strong (positive $NAM_{10}$





anomaly) or weak (negative $NAM_{10}$ anomaly). While the surface impacts of stratospheric extremes in individual winters has received much attention (Baldwin and Dunkerton, 2001; Domeisen, 2019; Charlton-Perez et al., 2018), the surface impacts of

consecutive sets of persistently anomalous winters have been less well studied and teleconnections between the stratospheric vortex and ocean variability are not well characterised or understood.

We examine the AMOC response to long-term variations in the stratospheric polar vortex using composite analysis of the AMOC strength following persistent anomalous $NAM_{10}$ intervals. We find oscillatory responses in the NAO and the AMOC consistent with previous work (Reichler et al., 2012). Diagnosis of the model supports a mechanism in which a persistently

strong vortex (positive NAM) perturbs a positive NAO anomaly which subsequently induces a positive AMOC response at 2-3 year lags via an increase in subpolar North Atlantic ocean-atmosphere heat flux. This, in turn, feeds back onto the NAO to drive a reversal in NAO phase (to negative) which leads to a subsequent negative AMOC anomaly at 15-20 year lags. The integrated effect of long-term oscillatory signals in the $NAM_{10}$ and the associated NAO variations may thus act as a metronome for the AMOC, which is a natural mode of oscillation in ocean circulation that varies on similar timescales. Further diagnosis of the

separate impacts from intervals of persistent positive NAM (strong vortex) and persistent negative NAM (weak vortex with repeated SSW occurrences) showed that persistent strong vortex intervals had a much larger impact on the AMOC, perhaps not surprisingly because the vortex anomaly is consistently present throughout the whole winter.

We additionally found prominent non-stationary variations across multiple timescales in the AMOC, NAO and $NAM_{10}$. Wavelet analysis revealed extended intervals in which 30, 50 and 90-year periodicities were dominant, so the composite re-

sponse patterns were complicated by the superposition of contributions from intervals exhibiting different timescale behaviour.

Interestingly, while all three indices ($NAM_{10}$, NAO and AMOC) co-varied at the 30 and 50 yr periodicities, only the $NAM_{10}$ and AMOC co-varied at the 90yr periodicity and co-spectra analysis suggested that the AMOC leads the vortex signal. This suggests a feedback of the AMOC variability onto the vortex that does not involve the NAO. A recent study (DM21) using the same UKESM simulation found long-term (90yr) co-variability between the vortex and the QBO. This suggests the possibility

that the 90yr AMOC-vortex relationship could act via an influence of the AMOC on equatorial wave forcing of the QBO, which then influences the vortex through the well-known Holton-Tan relationship. This was explored through wavelet and co-spectra analysis of variations in tropical east Pacific deep convection and QBO amplitude. These showed similar 90yr co-variability, indicating this as a plausible mechanism for the source of long-term variability in the QBO and the vortex, but further analysis of individual steps in the process, such as amplitude modulation of the various tropical waves that give rise to the QBO, would

be required to confirm this.

Finally, we have applied the model results, that link a lagged AMOC response to the presence of persistent vortex anomalies, to assess the possible contribution of the interval of persistently strong vortex in the 1990s to the recent observed changes in the AMOC. Our analysis suggests a maximum AMOC response at 50N with a lag of approximately 17 years. The lagged vortex-AMOC relationship is statistically significant using the AMOC response at both 50N ($r = -0.908$) and also at 30N

($r = -0.652$), the latter being closer in latitude to the RAPID Array observations. Using a regression technique we estimate that -0.49Sv of the observed RAPID Array AMOC trend by 2012 can be associated with the interval of persistent strong vortex behaviour centred on 1995. This represents nearly 17% of the total decrease in AMOC transport between 2005 and 2013 in the




RAPID array data. The observed negative trend in the AMOC has been attributed to a range of factors, including the influence of anthropogenic forcings (Caesar et al., 2018, 2021), but to our knowledge, the potential role of vortex variability has not

previously been considered. The origin of the interval of persistently strong vortex in the 1990s is unknown, but it was most likely due to internal variability. There is currently no consensus amongst climate models on how the vortex will respond to anthropogenic climate change (Ayarzagüena et al., 2020), and the 1990s appear to have been an anomalous period with no clear long-term trend emerging. As a result, our findings may indicate a significant role for internally generated signals in the recent negative AMOC trend.

There are several caveats to these results. Firstly, the results come from a single model, although there has also been limited analysis of vortex-AMOC interactions in CMIP5 models (Reichler et al. (2012). On the other hand, there is evidence that GCMs under-represent the influence of SSW events on the mid-latitude tropospheric jet, the NAO and surface temperatures, part of the "signal to noise" problem identified by Scaife and Smith (2018). Nevertheless, the model showed a clear NAO signal up to 3 months following vortex anomalies, indicating a reasonable representation of stratosphere - surface interaction. A previous

study using the UKESM (DM21) also noted an underestimation of SSW frequency compared to the ERA-Interim dataset, which indicates a positive bias in the mean vortex strength as well as the $NAM_{10}$, leading to the possible over-representation of positive $NAM_{10}$ intervals. The variability in the AMOC may also be under-represented in the simulation, as Roberts et al. (2014) suggests that the models' decadal variability is smaller than that in the RAPID dataset. Additional analysis using a suite of CMIP6 models would clearly be useful to assess the robustness of the results presented here.

Composite and wavelet analysis of time series data presented here is effective for identifying co-variability between the stratosphere and ocean. However, these techniques are less capable of demonstrating causality within modes of variability and so additional targeted experiments would be useful to establish this. Nevertheless, we have suggested physical pathways for vortex-ocean interactions on multi-decadal timescales which rely on well-established teleconnections (e.g. the in-season vortex-NAO connection) and demonstrate a possible key role for these interactions in recent AMOC behaviour. Our results

stress the role and importance of non-stationary signals for understanding long-term variability of the climate system. This complexity can be detrimental to analysis which relies on common stationary methods such as Fourier analysis. As a result, improved understanding and diagnosis of non-stationary climate variations as well as their underlying mechanisms will be key to overcoming these difficulties.



*Data availability.* ERA5 reanalysis data are available from the Copernicus Climate Change Service Climate Data Store

(CDS, https://climate.copernicus.eu/climate-reanalysis, C3S, 2017, last access: 10 August 2021). Data from the UKESM
simulation used in this study are available from the Earth System Grid Federation of the Centre for Environmental Data
Analysis (ESGF-CEDA;https://esgf-index1.ceda.ac.uk/projects/cmip6-ceda/,WRP,2019, last access: 6 August 2021).

*Author contributions.* OBDM conducted the analyses, and LJG, SO, JR, RS and BS directed the research. All authors were
fully involved in preparing and revising the text.

*Competing interests.* The authors declare that they have no conflict of interest.

*Financial support.* This research has been supported by the Natural Environment Research Council (grant no.
NE/L002612/1). Lesley Gray and Scott Osprey also receive funding from the UK Natural Environment Research Council
(NERC) through the National Centre for Atmospheric Science (NCAS) ACSIS project (grant no. NE/N018001/1) and the
NERC Belmont-Forum grant GOTHAM (grant no. NE/P006779/1). OBDM received support from the Oxford DTP in

Environmental Research (grant no. NE/L002612/1). JR was supported by NERC through NCAS, and through the NERC
ACSIS project (NE/N018001/1), and the UKRI-NERC WISHBONE (NE/T013516/1) and SNAP-DRAGON (NE/T013494/1)
projects. BS was supported by NOC ACSIS project (NE/N018044/1).

*Acknowledgements.* The authors would like to thank their respective funding bodies as well as Tim Woollings and Chris
O'Reilly for useful discussions. We also give thanks to the UKESM1 team who have worked to develop and run the model

used in this study as well as make the data available. In particular, Colin Jones, Jeremy Walton, Alistair Sellar and Till
Kuhlbrodt.



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




**Appendix A**

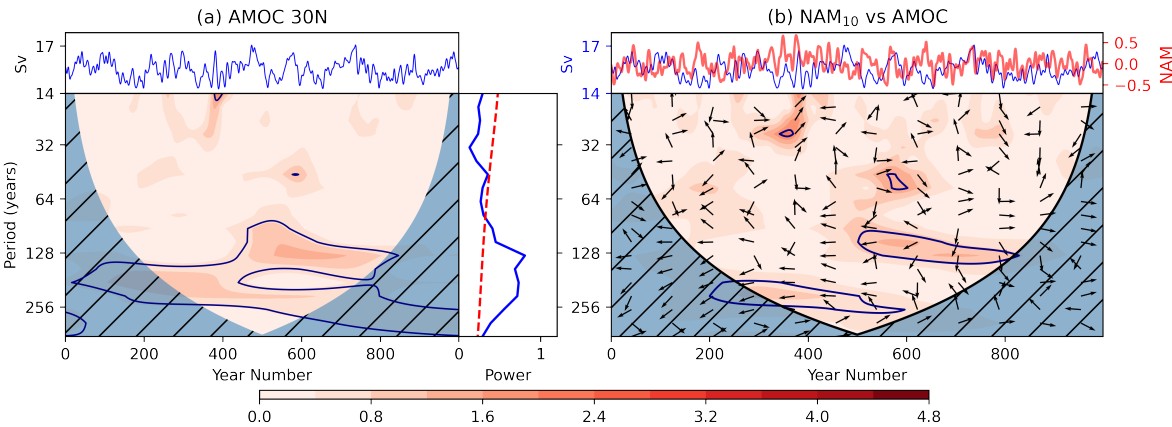

**Figure A1.** like figure 7 for the AMOC at 30N. **a** shows the wavelet power spectrum of the NAO and **b** the cross power spectrum between the AMOC and the $NAM_{10}$ index.

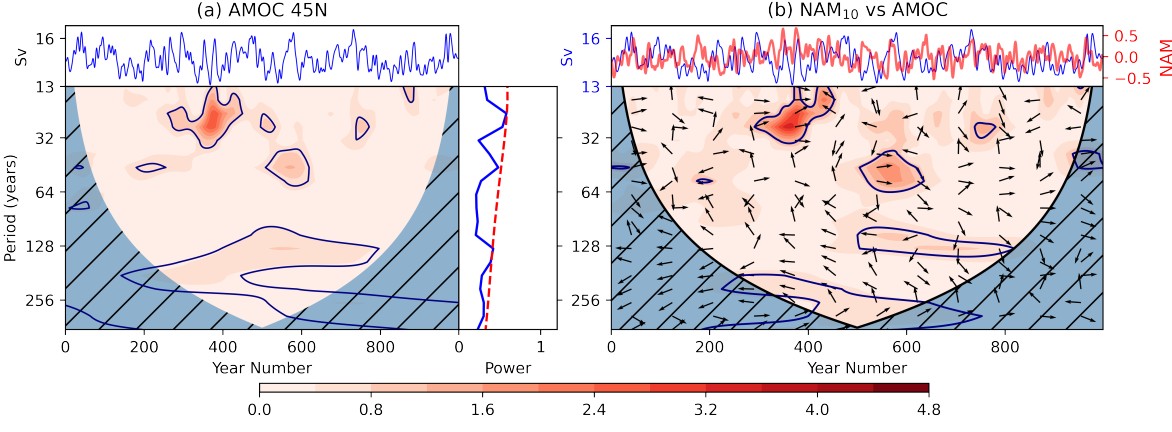

**Figure A2.** like figure 7 for the AMOC at 45N. **a** shows the wavelet power spectrum of the NAO and **b** the cross power spectrum between the AMOC and the $NAM_{10}$ index.






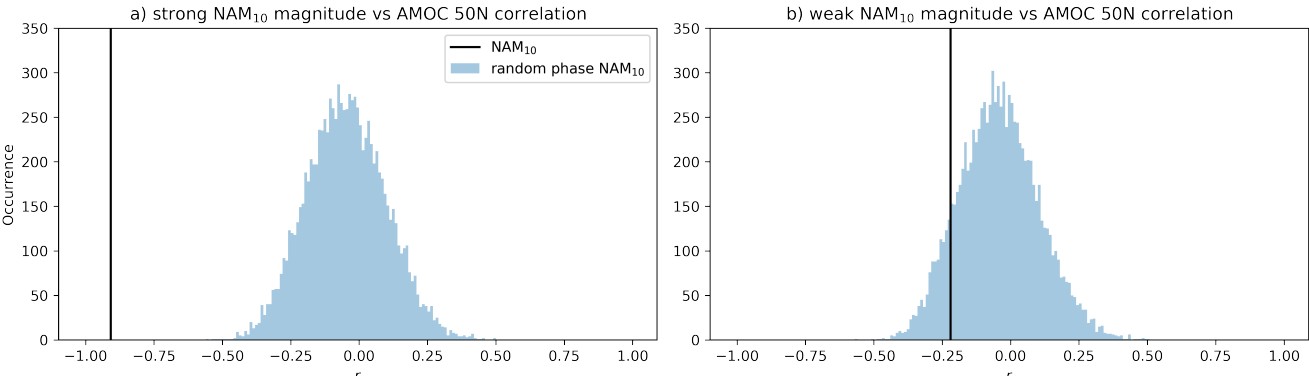

**Figure A3.** Probability distribution functions (PDFs) for correlations between the magnitude of $NAM_{10}$ extreme positive (**a**) and negative (**b**) values from surrogate $NAM_{10}$ data and anomalies in the AMOC at 50N evaluated 17 years later. Each $NAM_{10}$ surrogate is generated by applying a Fourier transform the smoothed $NAM_{10}$ index, randomly shuffling the Fourier phases and inverse transforming. Each PDF is built with 10000 surrogates and the correlation between the $NAM_{10}$ extreme magnitude and 17-year lagged AMOC anomaly at 50N is shown by vertical black lines in both subfigures.