# Peer review of "Interactions between the stratospheric polar vortex and Atlantic circulation on seasonal to multi-decadal timescales"

_Atmospheric Chemistry and Physics, 2021_

## Referee Comment (RC2)

**Review of the manuscript "Interactions between the stratospheric polar vortex and Atlantic circulation on seasonal to multi-decadal timescales" by Dimdore-Miles et al.**

The present manuscript analyzes the potential link between persistent extreme variations of the polar vortex and changes in the AMOC in a 1000-yr pi-control run of the UKESM1 model. The authors find an oscillatory response in the NAO and the AMOC after persistent extreme polar vortex intervals. Non-stationary variations of the polar vortex and AMOC at periods of 30 and 50 years seem to be related to these responses. However, AMOC variations at longer time periods (90 years) are found to precede changes in the polar vortex through modifications in the equatorial Pacific and the QBO.

The manuscript is well written, and the authors perform a thorough statistical analysis to identify the different non-stationary signals in AMOC, NAO and polar vortex and more importantly, the link between them. The topic is original as it has only been explored in a few previous studies. However, there are some aspects (most of them minor) that I think should be addressed. They are indicated below line by line.

L128: I think a ":" instead of "a" should be included after DM21.

L207-209: Baldwin and Dunkerton (2001) do not compute the NAM based on zonal mean geopotential height anomalies. It is Baldwin and Thompson (2009) who do that.

L213: It is written that after the central date of a $NAM_{10}$ extreme event, the $NAM_{10}$ must recover to westerly for a certain number of days. However, the $NAM_{10}$ is not a measure of the wind field. I would encourage clarifying this aspect of the definition.

L263: We → we

L281-282: I would not describe the SLP pattern for the -1-0 months before extreme $NAM_{10}$ events as a positive phase of the NAO. Strong centers of action in the Pacific and Siberian regions. In contrast, the anomalies over the Atlantic are very weak.

L308: I think a reference for the mechanism in which the SSTs respond to anomalous ocean-atmosphere heat fluxes is needed.

L309 and bottom row of Figure 1: Since a reference to the PDO is included in the text, I would recommend showing the whole Pacific in SST plots of Figure 1.

L371: Figure 4b

L373: Figure 4c "shows"

L386: I would change the title section and include something about the non-linearity of the response.

Figure 5: Figure caption and plots do not match. Panels' titles should also be corrected.

L414-416: The double peak in the sub-polar North Atlantic heat flux response to persistent polar vortex events is not that clear as in the AMOC.

L526-545: The authors relate the 90-yr vortex variability to a variability of the similar period in the QBO through a driving mechanism involving changes in the Pacific SSTs. However, no signal is detected in that field but in the East Pacific OLR. Can the authors explain this discrepancy?

L563: (Manney et al., 2005).

L582: Figure 11b.

---

## Author Response (AR1)

Response to Reviewer Comments

"Interactions between the stratospheric polar vortex and Atlantic circulation on seasonal to multi-decadal timescales" by Oscar Dimdore-Miles et al.

We thank the reviewers for providing their set of comments on our analysis. Their suggestions have helped us clear up technical points and more clearly demonstrate the proposed physical pathways in various. Below is a summary of the relevant changes made to the manuscript.

L128: I think a ":" instead of "a" should be included after DM21.
This has been changed to include a ':'.

L207-209: Baldwin and Dunkerton (2001) do not compute the NAM based on zonal mean geopotential height anomalies. It is Baldwin and Thompson (2009) who do that.
We have changed the text to clarify this, while Baldwin and Dunkerton used the NAM to measure the downwards propagation of vortex anomalies, we utilise the definition in Baldwin and Thompson 2009 (line 210).

L213: It is written that after the central date of a NAM10 extreme event, the NAM10 must recover to westerly for a certain number of days. However, the NAM10 is not a measure of the wind field. I would encourage clarifying this aspect of the definition.
This was an erroneous description of the definition and has been rectified (line 216).

L263: We → we
This has been changed.

L281-282: I would not describe the SLP pattern for the -1-0 months before extreme NAM10 events as a positive phase of the NAO. Strong centers of action in the Pacific and Siberian regions. In contrast, the anomalies over the Atlantic are very weak.
References to a positive NAO pattern preceding the stratospheric anomaly have been removed. We have included reference to the Pacific and Siberian centers.

L308: I think a reference for the mechanism in which the SSTs respond to anomalous ocean-atmosphere heat fluxes is needed.
We have added reference to Hausmann et al. 2017 which discusses mechanisms of heat flux driven SST variations in the Atlantic.

L309 and bottom row of Figure 1: Since a reference to the PDO is included in the text, I would recommend showing the whole Pacific in SST plots of Figure 1.
Figure 1 has been extended to show all latitudes and longitudes.

L371: Figure 4b
This has been added

L373: Figure 4c "shows"

**This has been added**

L386: I would change the title section and include something about the non-linearity of the response.

**This has been included.**

Figure 5: Figure caption and plots do not match. Panels' titles should also be corrected.

**This has been rectified so that Panels Titles and captions match.**

L414-416: The double peak in the sub-polar North Atlantic heat flux response to persistent polar vortex events is not that clear as in the AMOC.

This is a good point and remains one of the ongoing issues with the results. One possible explanation is that strong and weak vortex events are associated with perturbations in more complex spatial patterns of ocean-atmosphere heat fluxes (as opposed to a simple box average as we have used). To test this, we also measure the response of a more sophisticated Ocean-Atmosphere heat flux metric that is calculated by projecting the winter-time NA heat flux onto the spatial response pattern (strong – weak) to individual strong and weak vortex events averaged 0-60 days after the vortex anomaly. This index captures the time variation of the spatial loading pattern. The responses of this metric to different vortex interval types share some key features of the AMOC and indicates that the pattern of ocean-atmosphere heat flux associated with each interval type may consist of a more complex spatial structure than the sub-polar NA box.

We have added a panel in figure 5 for the responses of this new metric to different interval types (figure 5e), some new text describing it (see page/line number), and provided the loading pattern of ocean-atmosphere heat flux in figure A1.

L526-545: The authors relate the 90-yr vortex variability to a variability of the similar period in the QBO through a driving mechanism involving changes in the Pacific SSTs. However, no signal is detected in that field but in the East Pacific OLR. Can the authors explain this discrepancy?

Again, this is a fair point – one would expect a similar SST signal. A possible explanation is that the SST signal originates over a different Pacific region which projects onto the East Pacific OLR (and deep convection).

An analysis of the correlation between the AMOC at 50N and equatorial Pacific SSTs lagged by 45 years (in line with the pi/2 out of phase signals in OLR and AMOC on the 90-year timescale indicated by figure 9F) show various significant correlations centred over the central Pacific as opposed to the Eastern Pacific. Using this correlation map as a loading pattern, we are able to define an index of Pacific SST variation that tracks the temporal evolution of that spatial pattern. Furthermore, the cross-wavelet spectra between this metric and the east pacific OLR as well as the AMOC shows some co-variation on the 90-100 year timescale.

This extra cross spectrum has been added to figure 9 as well as a discussion of its potential importance in accounting for the OLR and SST signals. How these SST signals may influence the OLR in the east region is yet to be fully understood and the phase relationship between the series is unclear. Further study into this interaction is warranted but is outside the scope of this study. We have also added the correlation map (figure 10).

L563: (Manney et al., 2005).
**We have added parentheses.**

L582: Figure 11b.
**This has been changed**

1. For many places throughout the article, for example P12 L332 and L334, 15-20 years are suggested as the lag-time. But from the Figure 3 â4 and 5, it shows the significance response from 10-25 years ï¼or approx. 10-23 yrsï¼, of course strongest response is between 15-20 yrs. Maybe the authors could make the statements more accurate.

**We have amended the text to clarify that, while significant AMOC responses are evident between 10- and 23-year lags, the largest anomalies appear between lags of 15 and 20 years.**

2. Figure 3: I suggest reversing the order of the three rows, corresponding to the flow of description. In the caption, "black dots" should be corrected as 'blue dots'.

**Both the order and figure caption have been changed.**

3. Figure 4 caption, the second line: add "(green)" after NAO index and add "black" after "ocean-atmosphere heat flux".

**This has been changed.**

4. Figure 5: the label of sub-figures and their statement in caption are wrong. Please correct them accordingly. The label of third subfigure should be "c)".

**This has been rectified.**

5. Figure 7: "yellow contours" should be "blue contours".

**This has been changed.**

6. Besides the specific comments and technique corrections above, one suggestion to the authors which might improve the paper: to add one schematic diagram at the conclusion section to summarize the mechanisms: the relationship between NAM-NAO-AMOC (time scale: ~20 yrs ) as well as AMOC-Pacific deep convection(OLR)-QBO-NAM (time scale: ~90 yrs).

**While we appreciate that this may aid in showing the pathways involved, the majority of the mechanisms proposed in the paper are well known and comprehensively studied (e.g. the HT link as well as the vortex-NAO connection) so providing a diagram showing them may be superfluous**

given their coverage in previous literature. Perhaps more importantly, the timescales involved in this study (mainly 30-, 50- and 90-year periods) may be model dependant and other studies utilising different model data may see variability on different characteristic timescales. As a result, labelling a schematic with the three timescales found here may also be misleading.

References

Hausmann, U., Czaja, A. & Marshall, J. Mechanisms controlling the SST air-sea heat flux feedback and its dependence on spatial scale. *Clim Dyn* **48,** 1297–1307 (2017). https://doi.org/10.1007/s00382-016-3142-3

---

## Author Response (AR2)

Response to Corrections

"Interactions between the stratospheric polar vortex and Atlantic circulation on seasonal to multi-decadal timescales" by Oscar Dimdore-Miles et al.

We thank all the two reviewers as well as the handling editor for providing their set of technical corrections on our revised manuscript. We have addressed all points. We would also like to thank you for the help throughout the submission and review process.

Kind regards

The Authors